# Recycling fossil infrastructure for cleaner energy transitions

Hauke Schlesier [1,2], Gonzalo Guillén-Gosálbez [2] ✉ & Harald Desing [1] ✉

The climate crisis mandates building renewable energy infrastructure faster, increasing the demand for primary materials with large environmental footprints. Sourcing these materials from urban mines can mitigate such impacts, but the potential of recycling depends on waste availability. Here, we use life cycle assessment and monetization of impacts to explore the environmental implications of recycling fossil infrastructure that may become obsolete during the transition. We find that among many materials in fossil infrastructure, recycling steel and copper is particularly appealing, as their stocks (1.34 gigatons and 10.03 megatons) align with the projected energy transition demands (145% and 32% of median demand between 2020–2050, respectively). Recycling steel and copper in fossil infrastructure could save up to 1.95 gigatons $CO_{2,eq}$ and 11.69 trillion US Dollars in externality costs until 2050, while remaining competitive considering current production methods. Using recycled steel and copper would also reduce the carbon footprint of energy transition technologies—for example, wind and photovoltaic power—by one third.

Meeting climate goals will require deploying 6.0–8.1 terawatt peak (TWp) of wind and 8.5–14.0 TWp of solar photovoltaic (PV) capacity in 2050[1,2], for which metals involving mining and refining operations with large environmental[3–5] and social impacts will be needed[6,7]. Recycling bulk metals from waste streams—such as demolished buildings, end-of-life vehicles, industrial equipment, and consumer goods[8]—could reduce the demand for primary materials but the availability of waste is limited[9,10]. Besides avoiding impacts, building renewable infrastructure with recycled materials could accelerate the transition by circumventing the slow process of opening new mines (16.5 years on average[11]).

Previous studies showed that significant climate impact savings between 30% and 95% could be attained by recycling bulk metals compared to primary production[12–15]. However, the limited availability of metal waste and its competing uses (e.g., buildings, vehicles, packaging, machinery) makes it necessary to find additional metal waste sources to further enhance circularity and reduce impacts in the energy transition. We focus here on how the energy transition can be supported by using recycled metals from fossil infrastructure in clean energy systems, as renewable energy systems are expected to replace fossil ones, thereby making the resulting waste from decommissioning fossil infrastructure available for recycling purposes. Specifically, we study materials contained in fossil infrastructures such as coal mines, oil and gas rigs, pipelines, and power plants. Some of these infrastructures can be repurposed using circular strategies other than recycling, including repurposing boreholes of oil and gas wells for geothermal energy[16–19], converting oil-based refineries into bio-refineries[20], reusing pipelines for hydrogen transport[21], constructing grid infrastructure (e.g., large-scale battery storage) on former power plant sites[22], carbon storage in depleted oil and gas reservoirs[23], and producing synthetic fuels on offshore gas and oil platforms[20]. However, these options do not target the acceleration of building capacity in solar and wind, the dominant power sources in many transition plans[24–27]. Although previous work estimated the material stocks in fossil infrastructure in aggregated material categories[28,29], the feasibility and broad environmental implications of their recycling, especially for building clean energy systems, are not yet well understood.

[1]Technology and Society Laboratory, Empa—Swiss Federal Laboratories for Materials Science and Technology, St. Gallen, Switzerland. [2]Institute for Chemical and Bioengineering, Department of Chemistry and Applied Biosciences, ETH Zürich, Zürich, Switzerland. ✉e-mail: gonzalo.guillen.gosalbez@chem.ethz.ch; harald.desing@empa.ch

To fill this gap, we map global in-use stocks in the current fossil infrastructure for 22 materials, quantifying the recycling potential, and comparing the stocks of six materials with the estimated material demand induced by transition pathways. We find that steel is the largest metal stock in fossil infrastructure (1.34 gigatons (Gt)) and is within the range of estimated demand in transition pathways aiming at 1.5 °C and 2 °C heating[30]. Copper is another promising material to recycle (10.03 megatons (Mt) in fossil infrastructure), which could contribute around one third of the median energy transition demand. Using prospective life cycle assessment (LCA) and monetization factors (see Fig. 1 for method overview and Methods section for more details), we find that recycling steel and copper could save externality costs of more than 4.05 trillion USD$_{2023}$ (United States dollars in year 2023 equivalents), and decreases environmental impacts of steel and copper production in at least 17 out of 20 indicators—while remaining economically competitive.

## Results

### Fossil material stock estimate

We start by determining the amount of materials available in the current fossil infrastructure, finding that a total of 6.39 Gt (4.71–9.06 Gt), 95% confidence interval) of waste materials may become available through the transition. Concrete is the largest of these stocks, representing 2.01 Gt (1.24–3.42 Gt) and is mostly used for pavements and structural components. Yet, this stock is well below the estimated median demand induced by the energy transition of 6.49 Gt[30]. Moreover, recycling of concrete is limited as only aggregates can be replaced by crushed concrete, still requiring new cement, the main driver of climate impacts for concrete[31].

Steel is the second largest stock, amounting to 1.34 Gt (0.84–2.26 Gt), matching cumulative steel demand estimates for energy transitions including energy generation (solar, wind, nuclear, biomass, hydro, and geothermal energy), transmission, and distribution infrastructure from 2020 to 2050 (145% of the median demand, 0.93 Gt (0.51–2.85 Gt) depending on the scenario[30], see Fig. 2a; material demand of grid-scale batteries is excluded since battery capacity in the transport sector (electric vehicles) may be leveraged[30]). Steel is mostly used for structural purposes in oil and

gas extraction equipment (46%), power plants (26%), pipelines (21%), and coal mines (6%) (Supplementary Methods 1). The five countries with the largest fossil steel stocks are China, Saudi Arabia, Russia, the United States, and Iran, with 164, 155, 152, 150, and 94 Mt steel, respectively (see Fig. 3a), accounting for more than half of global stocks (53%).

Aluminum (Al) and copper (Cu) follow, used mainly in cables in fossil infrastructure (7.80 Mt (4.36–14.42 Mt) and 10.03 Mt (5.94–18.33 Mt) respectively), and could contribute up to 9% and 32% of the Al and Cu required in median projections of energy transitions[30]. In contrast, stocks of nickel (Ni) and silicon (Si) in fossil infrastructure are insignificant in comparison to the induced demand (0.06 Mt and 0.83 kilotons (kt), 4% and 0.001% of median demand for energy transitions until 2050 according to ref. 30; see Fig. 2a).

For some of the materials contained in current stocks, no demand for the energy transition was reported in ref. 30 (e.g., for Zinc (Zn); see Fig. 2b), or they are not typically contained in renewable energy infrastructure, such as barite; however, alternative uses could be explored. For example, 338.36 Mt (58.64–1847.04 Mt) of barite—used in drilling fluids and representing at least seven years of annual production—could be repurposed as radiative cooling paints containing barite[32], while 10.42 Mt (6.00–18.32 Mt) of stone wool in power plants could be reused for insulating buildings or district heating pipes.

Demand for cobalt is also not reported in ref. 30. The future demand is driven by battery deployment and projected to reach almost 500 kt a$^{-1}$ in 2050[33]. In fossil infrastructure, cobalt is used in superalloys for gas turbines and can be recovered by leaching superalloy scrap[34,35]. The total stock available in the current fossil infrastructure is 6.92 kt (median, see Fig. 2b), much lower than the expected cumulative demand for cobalt of 8.5 to 10.9 Mt from 2025 to 2050 (0.06% to 0.08% of demand). Therefore, recycling cobalt from fossil infrastructure might not play a major role, which is why we exclude it from further analysis.

We also exclude Al, Ni, and Si from further analysis, as recycling the stocks could at best contribute < 10% of the median demand generated by the energy transition (9%, 4%, and 0.001%, respectively). The potential of Al stocks in the energy transition is even lower when

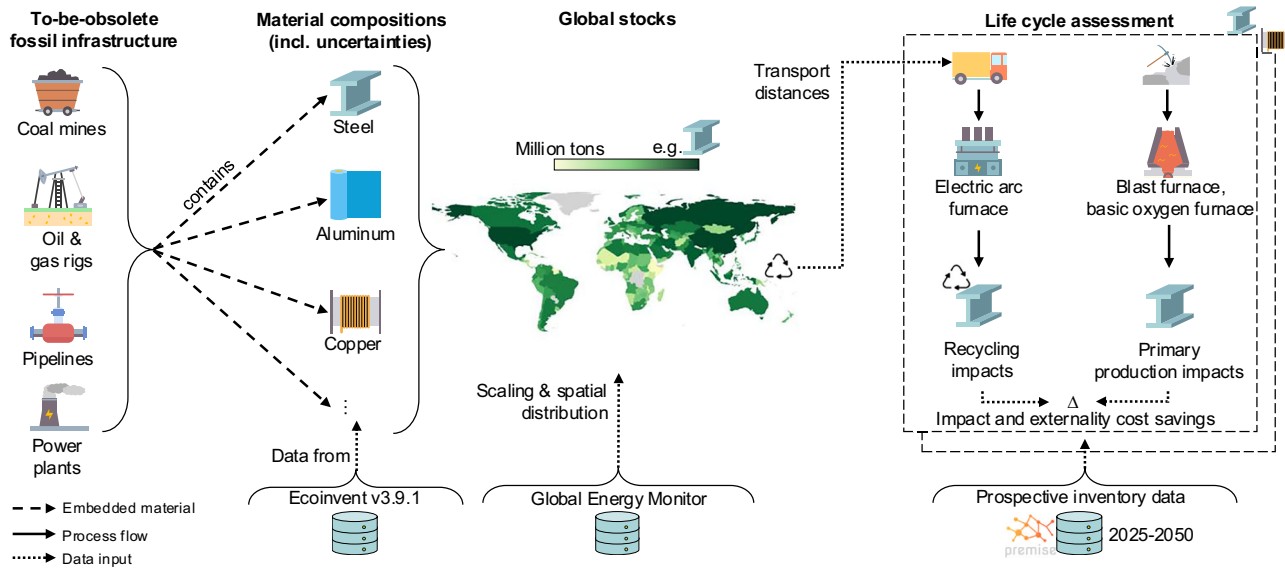

**Fig. 1 | Calculating the environmental impact and externality cost savings of recycling steel and copper in the current fossil infrastructure.** Global material in-use stocks of fossil infrastructure are determined by scaling material intensities (ecoinvent v3.9.1 database[75]) of coal mines, oil and gas rigs, oil and gas pipelines, and power plants with the size of global fossil infrastructure (according to Global Energy Monitor databases[64,65,70,76–78]). We then focus on steel and copper, estimating the impacts of primary and secondary production (using scrap from fossil infrastructure) for 2025 to 2050 (using Premise[39]), and translating them into monetary units using externality factors to quantify the benefits of recycling steel and copper for the energy transition.

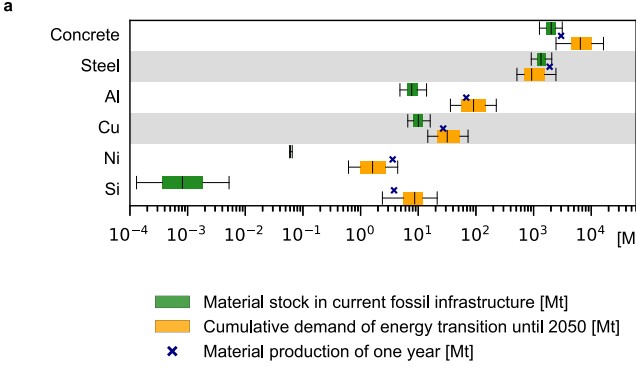

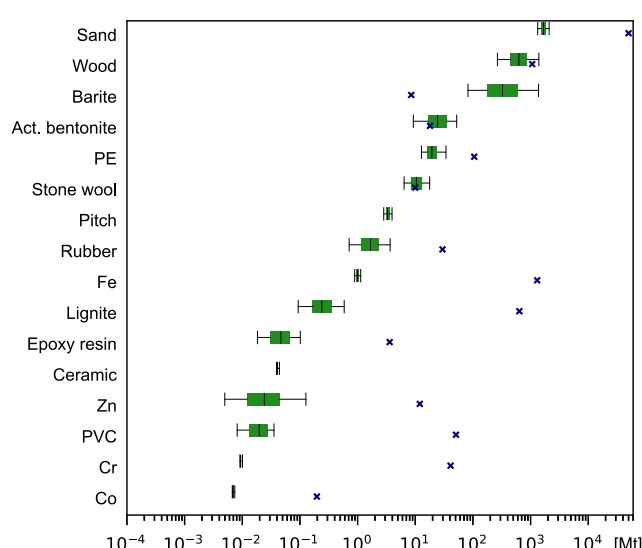

**Fig. 2 | The size of steel stocks in the current fossil infrastructure is of the same magnitude as required in energy transitions. a** Stocks of six materials stored in currently operating, mothballed, or idle fossil infrastructure (green), compared to cumulative demand for global energy transitions from 2020 to 2050 (excluding batteries; data from ref. 30 in yellow) and current annual production (blue cross, various sources, see Supplementary Table 7). Uncertainties (box shows the inter-quartile range, whiskers show 2.5th and 97.5th percentile of distribution, center line represents median; calculated using a Monte Carlo simulation with $n = 500$ random samples) stem from possible material intensities and linking of infrastructure types. The distribution of demands shows the range within possible material transition scenarios and models. Materials that are investigated in detail (steel, copper) are highlighted with a gray background. **b** Material stocks in current fossil infra-structure that are currently not in demand for energy transitions (or no demand reported in ref. 30, e.g., cobalt (Co)). Yet, they might become interesting when redesigning renewable infrastructure. Error bars (whiskers show 2.5th to 97.5th percentile, boxes capture the inter-quartile range, center line represents median) represent uncertainty regarding possible material intensities and linking options for infrastructure types and are calculated using a Monte Carlo simulation with n=500 runs. Al=aluminum, Cu=copper, Ni=nickel, Si=silicon, PE=polyethylene, Fe=iron, Zn=zinc, PVC=polyvinyl chloride, Cr=chromium, Mt=megatons.

considering that most clean energy applications require wrought-alloy Al (e.g., module frames, mounting systems, cables), which only accounts for 64% of the total Al in fossil infrastructure (thus only around 6% of the median energy transition demand of Al could be supplied).

We next focus on investigating the environmental savings of recycling steel and copper stocks. We first compare steel and copper stocks with the available recycling capacity to evaluate the feasibility of providing large amounts of secondary raw materials.

## Feasibility of recycling steel stocks and associated environmental impacts

The global capacity of electric arc furnaces (EAFs) and electric induction furnaces for steel recycling, with EAFs being today's dominant technology for steel recycling (97%), can process a maximum of 743 Mt a$^{-1}$ (ref. 36), yet EAF steel is produced at 541 Mt a$^{-1}$ globally[37], allowing for an additional 202 Mt of steel to be processed annually. The recycling capacity is expected to increase to more than 1500 Mt a$^{-1}$ until 2050 (in the Shared Socioeconomic Pathway 2 National Policies implemented (SSP2-NPi) scenario; Supplementary Methods 2), allowing for an idle capacity of more than 400 Mt a$^{-1}$ if capacity utilization remains the same. If the steel scrap from fossil infrastructure were released evenly across the period 2025–2050, the annual additional steel scrap volume would then utilize less than 40% of the global idle recycling capacity in any year (see Fig. 3a, SSP2-NPi, never exceeds 74% in any other scenario, Supplementary Methods 2), thus likely not posing a constraint on absorbing the additional steel waste.

Because environmental impacts from both secondary and primary production routes do change over time—e.g., through technological learning, changing energy mixes, or decreasing ore grades—we evaluate the environmental benefits of recycling over time (between 2025 and 2050) and for six scenarios. These future trajectories are generated by the Integrated Assessment Model (IAM) Remind[38] following SSP1, SSP2, and SSP5 in both NPi and Nationally Determined Contributions (NDC) variants. The life cycle inventory background database ecoinvent v3.9.1 is modified for each time step according to all IAM scenarios using Premise[39]. Projected global mean temperatures until 2100 in the investigated scenarios range from 1.9 °C in SSP1-NDC to 3.9 °C in SSP5-NPi (see Supplementary Methods 3), with SSP2-NDC (2.3 °C) and SSP2-NPi (3.1 °C) being close to the currently projected increase of 2.5 to 2.9 °C until the end of the century[40].

We find that during the period 2025–2050, recycling steel from fossil infrastructure would improve at least 17 out of the 20 LCA indicators across scenarios relative to the dominant primary production route (i.e., blast furnace and basic oxygen furnace; BF-BOF) (Fig. 3b for nine LCA indicators and Supplementary Figs. 18–19 for other indicators). More precisely, in the SSP2-NPi scenario, the largest impact reductions are found in metal depletion (92%), followed by particulate matter formation (72–78%), freshwater eutrophication (71–82%), and global warming potential (61–74%), with similar results for other SSP scenarios. In contrast, impacts from water use and ionizing radiation would increase slightly (5–8% and 14–23%, respectively). In general, the temporal evolution of the relative steel impacts (secondary versus primary production) follows a monotonic decrease in ozone depletion, climate impact, biodiversity loss, and marine eutrophication, driven by efficiency gains and continuing defossilization of supply chains. The relative land use impact of steel production is increasing, due to increased use of biomass in lubricating oil production; however, this can be avoided by switching to green chemicals synthesized from atmospheric carbon and water.

Environmental impact reductions of steel recycling are mainly driven by lower demand for coking coal, mining of coal and iron ore, as well as production of alloying metals, all of which cause significant climate impact and eutrophication, and particulate matter emissions. Blast oxygen furnace slag contributes to human and ecosystem toxicity in primary steel production, and can be avoided in secondary production. Both higher water demand in the EAF route than in the BF-BOF route, and water consumption in hydroelectric power plants used for EAF electricity supply cause increased water use in secondary steel production. Substituting fossil energy in primary steel-making with electric energy in EAF generates a trade-off with ionizing radiation from nuclear power plant waste and Uranium production in the investigated scenarios (Supplementary Figs. 20–22). This trade-off can

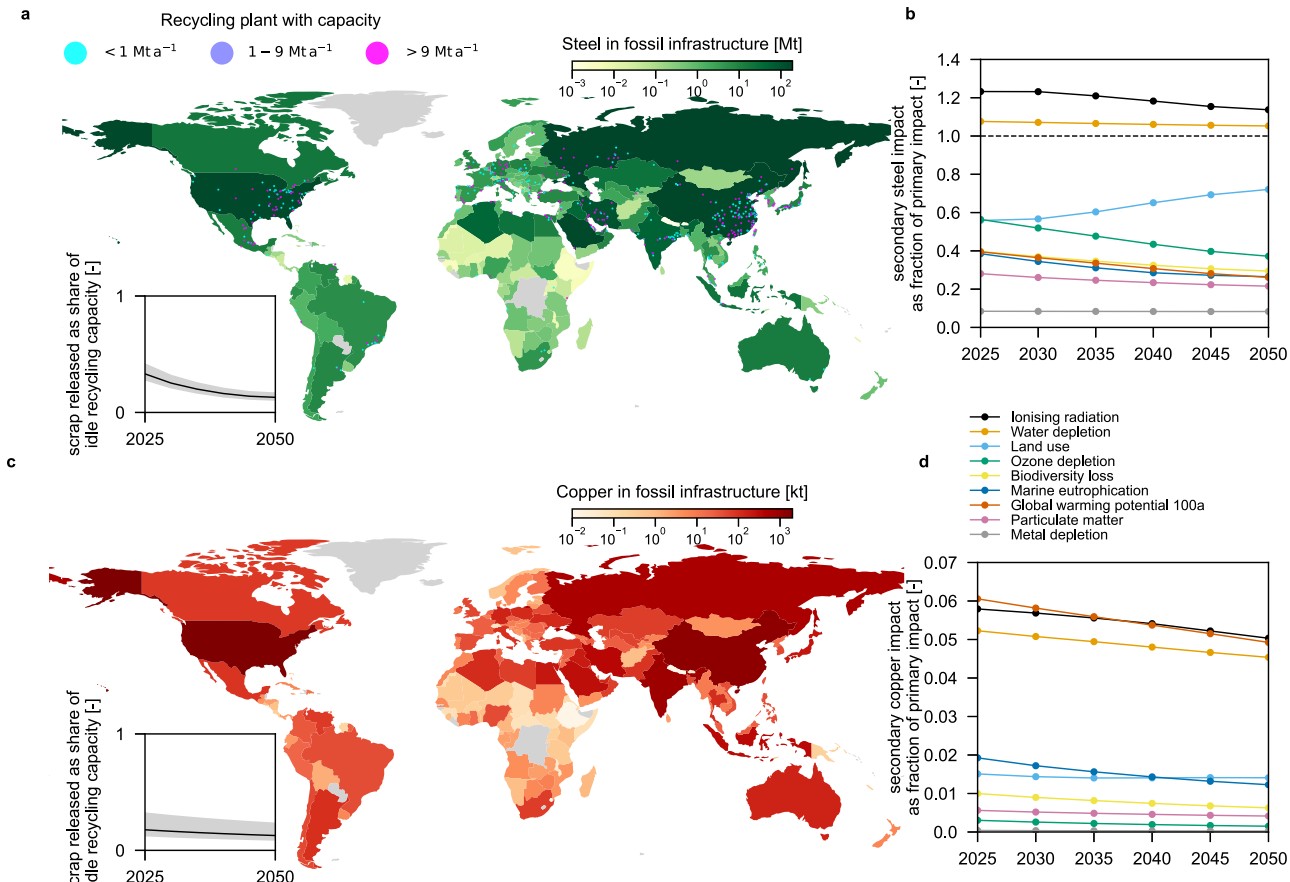

**Fig. 3 | Spatial distribution of steel and copper stocks in global fossil infrastructure. a** Green shades represent the magnitude of steel stored in fossil infrastructure in 200 countries. Gray color indicates countries or regions with missing data. Dots show the location and capacity of electric arc furnaces for steel recycling (data from ref. 36). The inserted axis shows global annual scrap release from fossil infrastructure as a share of global idle recycling capacity (scrap assumed to be released evenly between 2025 and 2050). Gray shades indicate the sensitivity of the share to the recycling capacity utilization (0.73 ± 10%; for details, see Supplementary Methods 2). Relative impact of recycling the fossil (**b**) steel and (**d**) copper stock compared to primary production across nine environmental indicators in the SSP2-NPi (Shared Socioeconomic Pathway 2 National Policies implemented) scenario; other 11 impact categories in Supplementary Figs. 18, 19, 23, and 24. **c** Red shades represent the magnitude of copper stocks in 200 countries. Gray shades in the inserted axis indicate sensitivity to the copper recycling capacity utilization (0.83 ± 10%). Mt=megatons, a=year, kt=kilotons.

be completely avoided when no nuclear power is present in the future electricity supply.

## Feasibility of recycling copper stocks and associated environmental impacts

Regarding copper, the secondary production capacity was 12.7 Mt a⁻¹ in 2023 and could reach more than 18.5 Mt a⁻¹ in 2050 under the current growth rate (Supplementary Methods 2). If the current copper stock in fossil infrastructure were recycled evenly until 2050, it would require less than 30% of the idle copper recycling capacity in any year and scenario (utilization factor of 0.83; see Fig. 3c). Overall, copper recycling capacities will be sufficient for absorbing the additional copper scrap from fossil infrastructure.

Copper recycling reduces environmental impact for all 20 LCA indicators when compared to primary production (Fig. 3d for nine LCA indicators and Supplementary Figs. 23 and 24 for other indicators). In the SSP2-NPi scenario, the largest reductions are found in metal depletion, particulate matter formation, ozone depletion, and biodiversity loss (>99%, see Fig. 3d). Reductions in marine eutrophication (98–99%), ionising radiation (94–95%), water use (95–96%), and global warming potential (94–95%) are only slightly smaller. These benefits arise mainly from avoiding copper mining and its sulfidic tailings—which release toxic compounds and drive eutrophication—as well as from avoided emissions of smelting (Supplementary Fig. 25). Over the period 2025–2050, the relative impacts of recycled copper decrease monotonically for 17 of the 20 indicators, primarily because declining ore grades make primary production progressively more burdensome (Supplementary Methods 4).

## Total environmental benefits

Scaling environmental benefits to the total steel and copper contained in fossil infrastructure (investigating a time frame of 2025–2050), between 1.68 and 1.95 Gt CO$_{2,eq}$ of emissions could be avoided depending on the scenario and year of intervention (>95% by steel recycling, see the Supplementary Figs. 18 and 23), which is comparable to Russia's greenhouse gas emissions in 2023[41]. Between 49 and 80 billion m²a of land could remain unoccupied from 2025 to 2050 due to the reduction in land required for steel and copper production (e.g., for mining, foresting), representing roughly the land occupation of Luxembourg[42]. Between 5.0 and 5.4 Mt PM$_{2.5,eq}$ particulate matter emissions could be avoided, equivalent to around three years of PM$_{2.5}$ emissions in the United States in 2024[43]. In regard to biodiversity loss, more than 9600 species years could be saved.

Testing the robustness of our results with pessimistic assumptions regarding dilution of steel scrap with added primary metals to compensate for lower scrap quality and higher hexavalent chromium leaching from waste steel slag and dust, environmental impacts can still be saved, though at reduced levels. E.g. CO$_{2,eq}$ savings from steel

recycling would be halved, and avoided particulate matter emissions decline from 3.35 Mt PM$_{2.5,eq}$ down to less than 1.85 Mt PM$_{2.5,eq}$. However, steel recycling remains the ecologically preferable option for all indicators except for ionizing radiation and water use (Supplementary Figs. 19–22), where impacts can become slightly larger for recycling if consumed electricity is partly produced with nuclear and hydropower. Also, human toxicity of secondary steel production can become larger than that of primary production due to increased levels of hexavalent chromium emissions (see the Supplementary Figs. 20–22), which can be reduced with better treatments of EAF slag and dust[44–47].

We also test pessimistic assumptions for copper recycling, accounting for low-grade scrap entering an additional smelter step instead of directly going to fire-refining and casting (see Supplementary Methods 4 for details). When compared to recycling high-grade copper scrap, we find that copper recycling remains the preferable strategy in all indicators, yet environmental savings would be reduced particularly for ionising radiation (20–21%, depending on the scenario), fossil depletion (17%), and climate impacts (16–17%). The effect on all other indicators is <15% compared to the high-grade scrap case (Supplementary Fig. 25).

### Externality costs
Saving environmental impacts also avoids future externality costs for human health (HH) and ecosystem damage (ED) remediation (see Fig. 4a), which are typically not accounted for in production costs. Reducing impacts via recycling thus also leads to healthcare cost savings due to reduced pollution, less compensation for impaired ecosystem services, and lower demand for natural resources (see Methods section for details). Recycling fossil steel and copper stocks could avoid externality costs between 4.10 and 6.94 trillion USD$_{2023}$ (Fig. 4a, SSP2-NPi scenario, 75–78% from steel recycling; between 4.05 and 11.69 trillion USD$_{2023}$ across investigated SSP scenarios, Supplementary Figs. 26 and 27), depending on the year of intervention (2025–2050). These cost savings are mainly realized by reducing

damages to HH caused by human toxicity and particulate matter formation, as well as climate change.

Avoided externality costs correspond to 1.08% to 3.92% of that year's gross domestic product (GDP) depending on the time of recycling and SSPs. Under pessimistic assumptions on EAF slag and dust treatment, externality cost savings from steel recycling can be realized in SSP1 and SSP2 scenarios, yet externality cost savings for steel diminish to zero in 'fossil-fueled developments' scenarios (SSP5; see Supplementary Figs. 26). The lowest externality costs of steel and copper production are found in the secondary route in 2025, with costs increasing 0.7–4.1% a$^{-1}$ or 8–221 USD$_{2023}$ a$^{-1}$ ton$^{-1}$ steel and 0.7–3.9% a$^{-1}$ or 4–275 USD$_{2023}$ a$^{-1}$ ton$^{-1}$ copper on average depending on the scenario (Supplementary Figs. 26 and 27).

The total costs of primary steel and copper production are dominated by externality costs (see Fig. 4b). Depending on the steel type (reinforcing, low-alloyed, or chromium steel; see Supplementary Methods 1 for split), externality costs can be as high as four to nine times the primary production costs, and even 14 times for copper (SSP2-NPi scenario). Recycling reduces externality costs, lowering the total costs of steel by 56–80% (depending on steel type) and by 97% for copper compared to primary production. Moreover, recycling could be economically competitive even when omitting externalities (−0.3 to −6.3% for recycled steel compared to primary route, plant average production cost depending on steel type[48]; −58% for recycled copper from cables; see Supplementary Methods 5), suggesting a win-win strategy to reduce impacts while diminishing production costs.

### Utilization in renewable energy systems
Recycling steel and copper from fossil infrastructure for the energy transition could lead to cleaner renewable energy technologies. For example, using recycled steel and copper for wind turbines allows reducing greenhouse gas emissions by 30.8% and 33.7% in on- and offshore systems respectively (in the year 2025, following an SSP2-NPi scenario, Fig. 5a), making externality costs of wind turbine

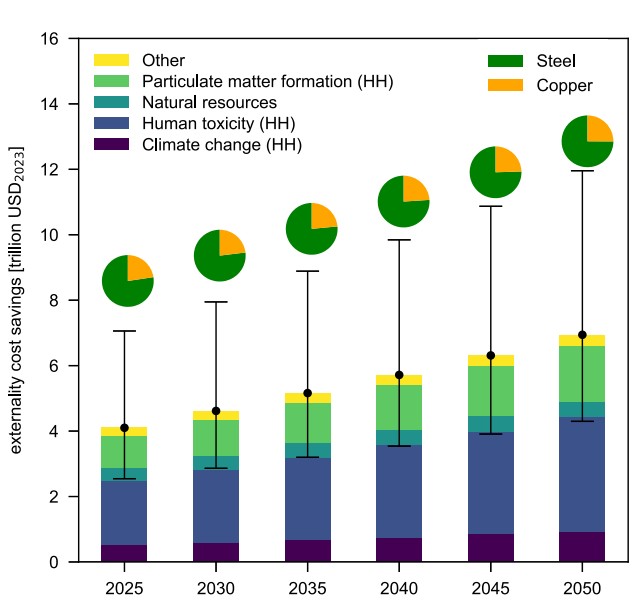
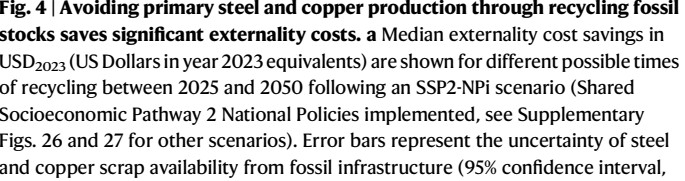
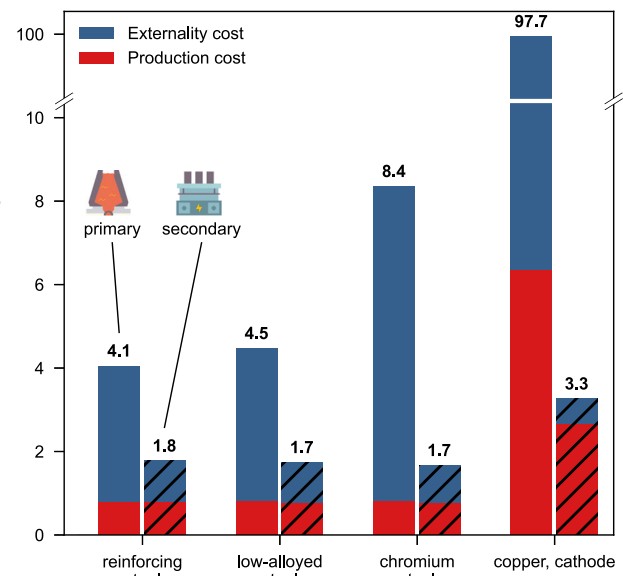

**Fig. 4 | Avoiding primary steel and copper production through recycling fossil stocks saves significant externality costs. a** Median externality cost savings in USD$_{2023}$ (US Dollars in year 2023 equivalents) are shown for different possible times of recycling between 2025 and 2050 following an SSP2-NPi scenario (Shared Socioeconomic Pathway 2 National Policies implemented, see Supplementary Figs. 26 and 27 for other scenarios). Error bars represent the uncertainty of steel and copper scrap availability from fossil infrastructure (95% confidence interval,

derived from Monte Carlo simulation with *n* = 500). Pie charts show the share of savings from steel and copper recycling. **b** Externality costs dominate total costs of primary steel and copper production (left bars) in 2025 in the SSP2-NPi scenario. Total costs (sum of production and externality costs) are more than halved through recycling (right bars) without increasing production costs. Variability of externality costs among scenarios in extended results (see Supplementary Figs. 26 and 27). HH=Human health, kg=kilogram.

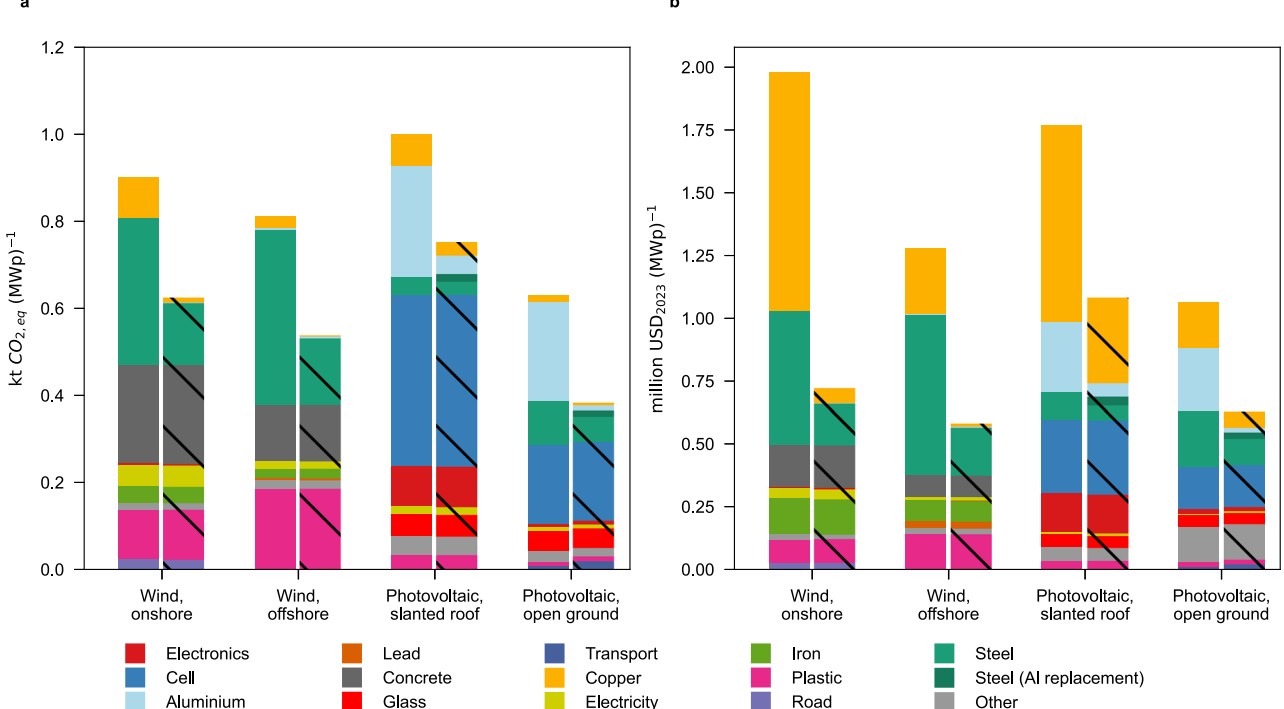

**Fig. 5 | Constructing wind turbines and photovoltaic (PV) systems with recycled steel and copper from fossil infrastructure reduces the climate impacts of renewable electricity. a** Greenhouse gas emissions from constructing low-carbon energy systems with primary steel and copper (wind turbines) or a market mix of aluminum (Al, in PV mounting systems) and copper are shown in the left bars, compared to construction with recycled steel and copper from fossil infrastructure in the right bars (hatched areas). MWp=megawatt peak capacity, kt=kilotons. Wind turbine inventories represent global weighted average, open-ground PV systems represent the European weighted average, and slanted-roof PV systems represent the Rest of World weighted average in 2025 following an SSP2-NPi scenario (Shared Socioeconomic Pathway 2 National Policies implemented). **b** Externality costs in USD$_{2023}$ (US Dollars in year 2023 equivalents) for constructing energy systems with primary steel and copper for wind turbines and market mix of aluminum for PV mounting systems (left bars) compared to recycled steel and copper used in wind turbines and PV mounting systems (right bars).

**Table 1 | Capacity potential of clean energy technologies that could be built with steel and copper recycled from the current fossil infrastructure (median stock) in comparison to the projected installed wind and solar energy capacity in the SSP2-NPi (Shared Socioeconomic Pathway 2 National Policies implemented) scenario and International Renewable Energy Agency targets[1,2] (year 2050; missing data for electrolyzers and transmission network)**

| Technology | Capacity potential | | | | |
| --- | --- | --- | --- | --- | --- |
| | **Steel** | **Copper** | **SSP2-NPi 2050** | **Climate goal** | **Unit** |
| Open-ground solar PV | 24.4 | 3.0 | 9.6 | 8.5–14.0 | TWp |
| Rooftop solar PV | 45.1 | 1.0 | 9.6 | 8.5–14.0 | TWp |
| Onshore wind | 10.0 | 3.7 | 5.1 | 6.0–8.1 | TWp |
| Offshore wind | 8.8 | 1.0 | 5.1 | 6.0–8.1 | TWp |
| Electrolyzers | 31.2–123.4 | 16.3–28.7 | - | - | TWe |
| Transmission network | $1.59 \times 10^9$ | $3.48 \times 10^6$ | - | - | km |

Intervals in the capacity potential column indicate range for different types of electrolyzer. The transmission network is assumed to be constructed for the medium voltage level. For steel and copper intensities of technologies, see the Supplementary Table 8. TWp=terawatt peak capacity, TWe=terawatt electrical capacity, km=kilometer, PV=photovoltaic.

construction drop by 63.5% and 54.5,% for onshore and offshore systems respectively (see Fig. 5b). Using all steel from obsolete fossil infrastructure to produce wind turbines would allow covering steel needs required for building 8.8 TWp or 10.0 TWp of offshore or onshore wind capacity respectively (see Table 1), i.e., 1.7 to 2.0 times the global wind energy capacity projected in Remind SSP2-NPi scenario in 2050 (and 1.1 to 1.7 times for reaching the targets of the International Renewable Energy Agency[1,2]).

Similarly, while steel is currently used in solar PV systems only for screws and hooks in rooftop installations and for support pylons in open-ground installations, it could replace commonly used Al for mounting systems[49] (for details see the Supplementary Methods 6).

Replacing Al in mounting systems for solar PV with secondary steel and using recycled copper in cables would curb greenhouse gas emissions by 39.2% and 24.8% for open-ground and slanted-roof PV system production, respectively. Furthermore, externality costs from open-ground and slanted-roof systems would be reduced by 40.8% and 38.9% respectively. Using the steel from fossil infrastructure for constructing PV systems would allow covering the requirements (assuming the aforementioned replacement of Al with steel) for building 24.4 TWp and 45.1 TWp of open-ground and rooftop PV, respectively. This is 2.5 to 4.7 times the projected installed PV capacity globally in the Remind SSP2-NPi scenario in the year 2050 (and 1.7 to 5.3 times the PV capacity in 2050 projected to reach climate goals[1,2]; Table 1).

The potential of building clean energy infrastructure using copper from fossil infrastructure is significantly less than for steel. More precisely, the copper stock is sufficiently large to supply between 10% and 31% of the installed solar PV capacity projected for the year 2050 following the SSP-NPi scenario, and between 20% and 72% of the projected wind capacity (Table 1). Yet, overall, recycling steel and copper from fossil infrastructure could contribute significantly to the steel and copper demand of the energy transition and lower the environmental impacts and externalities costs of transitioning to solar and wind energy.

Utilizing recycled steel and copper can also reduce climate impacts and externality costs of other relevant infrastructure, such as electrolyzers for green hydrogen production and power transmission infrastructure (see Supplementary Fig. 28; details on the inventories in the Supplementary Methods 4; we exclude batteries as their main application is expected to be primarily in the transport sector[30]). The alkaline electrolyzer shows the largest reduction for climate impact (38.8%) and externality cost (27.1%), both driven by recycled steel use. In contrast, benefits for power transmission infrastructure construction are driven by recycled copper use, leading to a 36.2% and 83.3% decrease in climate impact and externality cost, respectively.

## Discussion

Many countries are phasing out fossil fuels[50,51]—one recent example being the coal power phase-out in the United Kingdom[52]—or planning to do so in the future[51,53]. Recycling steel and copper from decommissioned fossil infrastructure could save significant externality costs for healthcare and adaptation to degrading ecosystems and worsening climate conditions. While adopting greener production routes is often hindered by higher costs[54–56], recycling steel and copper from fossil infrastructure could represent a win-win alternative.

Moreover, using recycled steel and copper for solar, wind, and other clean energy infrastructure could improve the environmental performance of entire value chains, such as green chemicals (e.g., green hydrogen), glass, non-ferrous metals, or even secondary steel itself. For example, the carbon footprint of green hydrogen, an energy carrier and fundamental molecule to activate the inert $CO_2$ in carbon capture and utilization routes[57], could drop by as much as 39%, making carbon capture and utilization routes much greener than originally thought (also reducing the footprint of other chemicals, e.g., green methanol, see Supplementary Discussion). We focus on the application of recycled materials in the energy sector, as fossil technologies need to be replaced with renewable ones, however, we clarify that the benefits of recycling materials can be realized regardless of the application sector as long as they displace primary production.

Furthermore, increasing metal waste availability by digesting fossil infrastructure may serve as a strategy to mitigate supply shortages for building renewable energy infrastructure[58], which might arise from geopolitical tensions and the concurrent need to accelerate climate action. For example, using recycled steel instead of Al—59% of which is smelted in China[59]—in mounting systems could facilitate a fast and resilient transition process.

Opposition to this strategy may arise from owners of fossil infrastructure—which are both state-owned and private companies—, who would prefer to prolong the lifetimes of their assets, despite the current regulations and agreements aimed at phasing out fossil fuels[53,60–62]. Reduced externalized costs, which are savings for society at large, could help to motivate the early retirement of state-owned fossil assets, potentially starting with the most polluting infrastructure[63]. In contrast, private owners have no direct benefits from reduced externality costs. In this case, regulation or incentives would be necessary to decommission infrastructure. The material value contained in the assets (around 537 billion $USD_{2023}$ in the case of steel and copper; see Supplementary Discussion) could further motivate to scrap them instead of leaving infrastructure mothballed or idle, as is currently the status for 210 power plants in 52 countries[64,65].

Resistance to retiring may vary depending on the type of infrastructure. For instance, coal power is becoming less economical compared to solar and wind energy[66–68], which might make power plant and coal mine owners more inclined to decommission their facilities. In contrast, oil and gas extraction continues to generate substantial financial profits[69], potentially making owners of oil and gas rigs less likely to decommission before the end of their lifespan. However, the average operating oil and gas rig is currently only six years away from the average retirement age[70]. Consequently, steel waste from oil and gas rigs might become available soon as they reach the end of their operational life.

Digesting fossil assets today would lead to lower externality costs and prevent them from being reactivated and causing further emissions. This has happened recently in Hungary, North Macedonia, and Italy[71], where coal-fired plants were converted into gas power plants. Early decommissioning would also prompt fossil firms to reorient their businesses, predominantly by investing more heavily in renewables. Early decommissioning would also have global implications on the supply and price of fossil electricity, fuels, and petrochemicals. Hence, a smooth transition should be sought by deploying alternative technologies at the right time and location (e.g., renewable power, electric cars, renewable carbon-based chemicals), so they would temporarily coexist with the fossil infrastructure[72].

Moreover, future dynamic models could capture systemic saving effects of potentially faster and cleaner energy transitions[72,73]. IAMs could also use our results to integrate externality costs for steel and copper production to enhance cost-benefit analysis[74]. Our quantitative analysis is intended to guide policy-making in the sustainable transition towards cleaner energy systems, where socioeconomic and geopolitical dynamics should also be considered when delineating future plans.

Our study is not exempt of limitations (see Supplementary Discussion), where the main ones are not including oil refineries and internal combustion engine vehicles in the potentially obsolete infrastructure assessed, and not including other strategies than recycling. Further, for simplicity, we do not focus on recycled materials' utilization in the transport sector (e.g., electric vehicles); nevertheless, the transport transition is also integral to the broader energy transition (e.g., energy storage in battery-electric vehicles). However, we still believe the main conclusions are robust and would hold in the face of uncertainties.

## Methods
### Material compositions
We take material intensities of fossil infrastructure from ecoinvent v3.9.1 cutoff[75], accounting for geographical variability, different sub-technologies, power plant classes, and fuel types as represented in ecoinvent. Intensities are normalized to capacity for coal mines and power plants, to length for pipelines, and cumulative output for oil and gas rigs. Uncertainties of material intensities are considered from (i) life cycle inventory (LCI) uncertainties in ecoinvent and (ii) the necessity of linking and matching infrastructure classes in ecoinvent versus the classes reported in Global Energy Monitor databases[64,65,70,76–78]. Borehead lengths in oil and gas extraction facilities are commonly not reported, though an important driver of material intensities. All these sources of uncertainty are propagated to the final stock estimates using a Monte Carlo approach with 500 runs. For more details, see the Supplementary Methods 7. We only consider on-site materials, thus neglecting grid connection and buildings, and excluding operational material flows, such as water and chemicals, as they likely do not accumulate in infrastructure stocks.

### Scaling to global stocks
To calculate total stocks, we consider currently operating, idle, or mothballed fossil infrastructure. Capacities that are already

decommissioned, currently under construction, or planned are excluded. Material stocks and their uncertainties are calculated individually for each infrastructure worldwide. Material intensities for each material are scaled with the size of each infrastructure taken from Global Energy Monitor databases[64,65,70,76–78] (for details, see Supplementary Methods 7). Data gaps occur for oil and gas wells and are filled by estimating cumulative production by multiplying the mean annual well production for a respective country by the well's years of operation. The capacity of coal mines—where missing—is calculated from annual production values assuming a capacity factor of 0.75 (based on the range of more than 0.5 to 0.9 depending on year and region[79]; total stocks are unsensitive to the capacity factor). Pipeline stocks are allocated to countries proportionally with the segment's length in the respective country. Pipeline segments in international waters are assigned to the nearest country.

## Life cycle assessment of steel
The functional unit is 1.34 Gt (0.84–2.26 Gt) of steel produced, and the geographical scopes are globally weighted averages in the years 2025, 2030, 2035, 2040, 2045, and 2050. LCIs are modeled for three types of steel—reinforcing steel, low-alloyed steel, and chromium steel—using adjusted data from Premise v2.2.3[39]. Premise is a Python package that translates the outputs of IAMs into changes in LCA background models, e.g., by adjusting for future energy mixes or efficiency changes of industrial technologies.

We model the primary and secondary production route with BF-BOFs and EAFs, respectively, as they are the by far dominant production technologies (61.3% and 20.2% of global steel-making capacity, with the alternative open hearth furnace only 0.6% and the rest a combination of the three)[80]. Primary production consists of iron ore extraction, conversion to iron sinter and pig iron in blast furnaces, steel production via basic oxygen furnaces, and dust, slag, and sludge treatment. The secondary production model includes the collection, sorting, and pressing of steel scrap from fossil infrastructure, transport, melting in EAFs, and dust and slag treatment. For more details on system boundaries applied, transport distances, and process efficiencies, see the Supplementary Methods 4.

Low-quality steel scrap needs to be diluted with primary sponge iron and alloying metals to compensate for contamination. If steel scrap of the same type and quality is recycled in an unmixed waste stream—as could be possible for large stocks such as pipelines or other fossil infrastructure stocks—, the need for dilution can be reduced. To reflect these two possibilities, we introduce pessimistic and optimistic recycling scenarios, assuming low and high scrap quality and slag composition, respectively.

Several adjustments are made to the LCI data from Premise: (1) Scrap supply is modeled reflecting country-weighted transport distances from fossil infrastructure to the closest EAF plant (assumed to be the most economic option). It is assumed that scrap is burden-free as it is collected at fossil infrastructure (same as in Premise databases). (2) Ecoinvent v3.9.1 likely overestimates hexavalent chromium ($Cr^{6+}$) contents in slag. We adjust $Cr^{6+}$ emissions to values found in the literature (see Supplementary Methods 4), selecting the lower values for the optimistic scenario and upper values for the pessimistic scenario.

## Life cycle assessment of copper
The functional unit is 10.03 Mt (5.94–18.33 Mt) of copper produced as the current globally weighted average and projected for the period 2025–2050. We model LCIs both for the pyrometallurgical (accounting for 80% of today's primary production[81]) and hydrometallurgical route (20%) for primary copper production and the electrorefining route for secondary production (see Supplementary Methods 4 for details).

Primary pyrometallurgical production consists of mining and beneficiation of copper ore concentrate, smelting, fire-refining,

casting, and electrorefining. Hydrometallurgical copper production includes mining, crushing, and grinding of copper ore, leaching, solvent extraction, and electrowinning. For secondary copper production, we model scrap collection, transport, fire-refining, casting, and the electrowinning process. We also include a pessimistic scenario for secondary copper production—assuming low-grade copper scrap—, where scrap is smelted to increase purity before being fire-refined. For more details on system boundaries applied, assumptions, and data sources for process inputs and outputs, see the Supplementary Methods 4.

## Impact and externality cost assessment
We evaluate 17 environmental midpoint indicators included in the ReCiPe 2016 impact assessment framework[82] (hierarchical perspective, e.g., only impact mechanisms based on scientific consensus and impacts within 100 years) and three additional environmental indicators: Climate impact based on global warming potentials from the Intergovernmental Panel on Climate Change (IPCC) 2021 over 100 years[83], land use, and biodiversity loss. For details, see the Supplementary Methods 8.

Environmental impacts are calculated by converting LCI flows—including raw materials, energy, and water inputs as well as emissions to water, soil, and air—into impacts using characterization factors as in equation (1).

$$\text{IMP}_{m,f,j} = \sum_k \text{LCI}_{k,f,j} \cdot \text{CF}_{k,m} \ \forall m \in M, \forall f \in \text{FU}, \forall j \in J \quad (1)$$

$\text{IMP}_{m,f,j}$ is the impact for indicator $m$ (belonging to the set of all disaggregated indicators $M$) associated with the functional unit $f$ (of all functional units FU, e.g., different steels) in year $j$ (belonging to the set of assessed years $J$). $\text{LCI}_{k,f,j}$ is the LCI input or output of the elementary flow $k$ associated with supplying $f$ in year $j$, and $\text{CF}_{k,m}$ is the characterization factor of $k$ for indicator $m$.

To calculate externality costs, we multiply the endpoint indicator impacts—given in the unit DALY (Disability Adjusted Life Years) for HH and species years lost for ED—with monetization factors as in equation (2):

$$\text{EXT}_{e,f,j} = \text{MF}_{e,j} \cdot \sum_{m \in M_e} \text{IMP}_{m,f,j} \ \forall e \in E, \forall f \in \text{FU}, \forall j \in J \quad (2)$$

Here, $\text{EXT}_{e,f,j}$ is the externality cost in $USD_{2023}$ generated in the endpoint externality class $e$ (HH or ED; $E$) while supplying the functional unit $f$ in year $j$. $\text{MF}_{e,j}$ is the monetization factor depending on the year $j$ and externality class $e$ (the cost of one unit of endpoint impact in HH or ED), and $m$ denotes the disaggregated ReCiPe indicators, where $M_e$ is the set of disaggregated endpoint indicators connected to the externality class $e$ ($M_e \subseteq M$). Note that the ReCiPe endpoint indicator impacts on natural resources are already calculated in USD.

## Monetization of impacts
To calculate the externality costs of environmental impacts, we follow the approach used in ref. 84. For the year 2018, the HH cost for one DALY is 176,624 $USD_{2018}$, with a lower and upper bound of 148,288 $USD_{2018}$ and 200,236 $USD_{2018}$, respectively[84]. The externality cost for one unit of species loss is 17,891,594 $USD_{2018}$, with a lower and upper bound of 4,472,899 $USD_{2018}$ and 44,728,985 $USD_{2018}$, respectively[84]. To estimate monetization factors in the future, we apply the benefit transfer concept[85] based on the approach followed in ref. 86 and as in equation (3).

$$\text{MF}_{e,j} = \text{MF}_{e,j'} \cdot \left(\frac{\text{GDP}_j}{\text{GDP}_{j'}}\right)^{\varepsilon_e} \ \forall e \in E, \forall j' = 2018, j \in J \quad (3)$$

Where GDP$_j$ is the gross domestic product per capita in year $j$, and $\varepsilon_e$ is the income elasticity for the willingness-to-pay for each externality class $e$, which is 1.10 and 0.38 for HH and ED, respectively[87,88]. GDP per capita values are calculated based on GDP and population prospects for respective SSP scenarios from 2025 to 2050 (data from ref. 89).

## Total cost estimation

The total cost comprises of production costs and externality costs. Whereas externality costs are outputs of our LCA, production costs for steel are taken from the Global Steel Cost Tracker (GSCT) database[48], which contains steel production costs for individual steel plants in 12 countries covering around 90% of global steel production. We link each steel plant to the steel products they produce (e.g., reinforcing steel, chromium steel) using data from the GSCT database[36] and then estimate the production costs for reinforcing, low-alloyed, and chromium steel individually, taking plant averages. For the estimated production cost of primary and secondary copper production, see Supplementary Methods 5.

## Reporting summary

Further information on research design is available in the Nature Portfolio Reporting Summary linked to this article.

## Data availability

The data generated in this study have been deposited in the Zenodo database under accession code 17361225 (https://doi.org/10.5281/zenodo.17361225). Size and location of fossil infrastructure is taken from Global Energy Monitor databases: https://globalenergymonitor.org. To calculate steel production costs, we use the Global Steel Cost Tracker (https://www.transitionzero.org/products/global-steel-cost-tracker). The material demand for energy transitions depicted in Fig. 2 are taken from Wang et al. (https://doi.org/10.1016/j.joule.2023.01.001). Source data are provided with this paper.

## Code availability

A code to reproduce the results is available at a Zenodo repository (https://doi.org/10.5281/zenodo.17361225).

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

## Acknowledgements

H.S. and H.D. were supported by the Swiss State Secretariat for Education, Research and Innovation (SERI) under contract number 22.00166 in the frame of Horizon Europe "CircEUlar: Developing circular pathways for an EU low-carbon transition" co-funded by the European Union (project 101056810).

## Author contributions

H.S., H.D., and G.G.G. designed the study. H.S. curated the data, carried out the modeling, created the figures, and wrote the first draft of the manuscript. H.S., H.D., and G.G.G. performed the analysis and contributed to editing and reviewing the manuscript.

## Competing interests

The authors declare no competing interests.
