## [Transparent Peer Review file · Nature Communications]

Recycling fossil infrastructure for cleaner energy transitions

Corresponding Author: Dr Harald Desing

Version 0:

Reviewer comments:

Reviewer #1

(Remarks to the Author)

This paper titled “Digesting fossil infrastructure for cleaner energy transitions”, analyzes the stock of 22 materials present in fossil infrastructure, including: mines, wells, refineries, and pipelines for coal, oil, and natural gas. It considers the demand for 6 materials of the energy transition (concrete, steel, aluminum, copper, nickel, and silicon) for electricity generation applications (photovoltaic and wind). After this analysis, it focuses on the benefits of steel recycling, as it finds that the stock from fossil infrastructure is of the same order of magnitude as the demand of steel for the energy transition. It considers important aspects for recycling, such as the global capacity of Electric Arc Furnaces (EAFs), the location of fossil infrastructure and recycling plants, and the approximate time needed to process this infrastructure. In addition, it compares the externality costs of recycling versus primary production.

The paper is very thorough and provides valuable information in the context of energy transition. Nevertheless, several aspects should be considered. In order of criticality:

First, the paper only considers wind and photovoltaic power as renewable infrastructure, overlooking a key aspect of the energy transition: storage. The main disadvantage of renewables compared to fossil fuels is that renewables consist of an intermittent flow, whereas fossil fuels are a stock. In other words, fossil fuels are available on demand, while renewables are not. The necessary solution is storage. The energy transition will not succeed without storage, and therefore this aspect should be considered. One way to store electricity is through batteries, and the most efficient type (Li-ion) contains significant amounts of cobalt. For this reason, the statement in lines 53–55 (“For some of the materials contained in current stocks, no demand for the energy transition was reported in [30] (e. g. for cobalt (Co); see Fig. 2b), or they are not typically contained in renewable energy infrastructure”) is surprising. Cobalt is often used in superalloys for gas turbines (see Figure S4), so this element could be recovered and reused in batteries. Moreover, cobalt is a conflict metal, critical, and very scarce in nature, so its inclusion is important.

Second, the paper only considers the recycling of steel, but why does it ignore the recycling of aluminum and copper? Figure 2a shows that the stock of copper and the demand for the energy transition are close. In addition, this metal is experiencing a significant decline in ore grade, which is causing increasing impacts on its extraction and could compromise its supply in the future. Furthermore, it is a strategic metal, as it is not only used in energy infrastructure but is also essential for digitalization. This paper would be much more complete if the recycling of these metals were included or it should be better justified why these metals have not been considered.

Third, the paper only considers electrical technologies but ignores hydrogen infrastructure. Green hydrogen (produced via electrolysis) is key to decarbonizing applications that are difficult to electrify (such as high-temperature heat). It is also critical for the production of e-fuels, as acknowledged by the authors in lines 180–182 (“For example, the carbon footprint of green hydrogen, an energy carrier and fundamental molecule to activate the inert CO₂ in carbon capture and utilization routes [60], could drop by as much as 30%”). However, this infrastructure is not considered in the study. Therefore, the authors should explain why these technologies are not included in the study.

Fourth, most of the steel corresponds to oil and gas extraction equipment (line 43). However, much of this infrastructure is located in very remote areas or consists of pipelines at some depth, which could make metal extraction challenging. How is the dismantling of this infrastructure being approached?

Fifth, to maximize the recycling of alloying metals in steel (some of which are critical, like Ni), it could be important to classify

them in order to obtain steel of similar quality. Has this aspect been considered?

Lastly, regarding lines 108–109 (“Between 25 and 45 billion m² of land could remain unoccupied”): does this figure account for the hectares that will be occupied by renewable technologies?

In conclusion, the title promises the digestion of fossil infrastructure for the energy transition but only focuses on electricity generation (wind and solar technologies) and steel recycling. Therefore, the authors should consider broadening the scope of the paper (including the recycling of more metals such as copper or cobalt and considering more technologies of the energy transition such as batteries or renewable hydrogen generation), or giving it a more specific title.

Reviewer #2

(Remarks to the Author)

General comment

Thanks for your invitation. There are several major logical gaps in this study that warrant careful attention.

First, while the study devotes significant effort to discussing recycled steel, its title focuses on “digesting fossil infrastructure,” which appears somewhat inconsistent with the article’s main content.

Second, recycled steel from decommissioned fossil infrastructure can be utilized across a variety of sectors, including construction and transportation, not necessarily in renewable infrastructure. Why, then, is the analysis limited to energy transition infrastructure? A clearer logical connection between the recycling of fossil infrastructure and the development of clean energy systems needs to be established.

Third, the study’s definition of energy transition infrastructure seems to include only wind turbines and photovoltaic systems—why are other forms of clean energy infrastructure excluded?

Fourth, even in the era of energy transition, it remains necessary to retain some fossil fuel infrastructure. Does this mean that the existing stock of fossil infrastructure does not need to be entirely “digested”?

Abstract

1)The first two sentences suggest that recycling can mitigate the environmental impacts of renewable energy infrastructure, which could make the readers think the topic of this study is about recycling renewable infrastructure. However, the focus of your study is actually on the recycling of fossil infrastructure. This creates a logical inconsistency that may confuse readers.

2)Regarding the 1.35 Gt steel stock mentioned in the abstract—what is its source? Does it refer to the in-use stock within fossil infrastructure to be obsolete? Additionally, where does the projected demand for the energy transition come from, and what is the time frame it covers? I think you need to clarify clearly the sources of material demand for energy transitions throughout the whole paper.

3)What is the time period over which the reported CO₂ emissions reductions and cost savings are calculated?

4)The final sentence references ‘socioeconomic and geopolitical issues,’ yet these aspects are not addressed or supported by the findings presented in this study.

Introduction

5)Line 26: the steel stock is (1.34 Gt) is a little different from the figure (1.35 Gt) in abstract.

6)Figure 1: what is ‘soon-to-be-obsolete’ fossil infrastructure? How to determine which infrastructure is about to be obsoleted?

Results

7)Line 43-46: specific figures or tables need to be depicted to support these results, which can not be seen from Figure 2.

8)Figure 2a: There are only 6 materials shown in Figure 2a, not 22.

9)Figure 2b: Again, you need to define clearly the material demand for energy transitions. Actually, cobalt is a critical material used in batteries, which are also important for energy transitions, but you didn’t take cobalt as an energy transition metal in this study.

10)Line 61: I think there needs a full explanation of why you chose steel to investigate the recycling benefits. The “relevance” here is too ambiguous.

11)Line 67-68: how did you conclude it?

12)Line 65: What does “Mt/a” mean—does it refer to “Mt per annum”? If so, comparing the ideal recycling capacity (in Mt/a) with the total steel stock is conceptually misleading, as the stock will not retire in a single year but over an extended period. A more appropriate comparison would be between the recycling capacity and the annual volume of steel retiring from fossil infrastructure.

13)Line 71-77: How did you tell “recycling of steel from fossil infrastructure would take five to 14 years with existing idle recycling capacity” from Fig. 3a? I don’t see any information about recycling time in Fig. 3a. In addition, the definition of recycling time needs to be explained cautiously because the recycling capacity can increase, which will shorten the recycling time. In addition, it is confused and odd that you evaluate the feasibility of recycling steel stocks through a comparison between the recycling time and energy transition time. I think a better story should be told here.

14)Figure 5: I understand that the aluminum is replaced by secondary steel so that reducing the carbon emissions related to aluminum. I suggest that you can separate the carbon emissions of steel in PV systems in the right bars (hatched areas) into two segments correspond to the recycled steel that replaces primary steel and the recycled steel that replaces aluminum, respectively.

15)Will it increase the installation costs by replacing Al with steel in PV mounting system? If so, the way replaces Al with steel maybe not economically viable in practice.

Discussion

Line 178-183: All the benefits are actually attributed to the recycled steel. It is obvious that using recycled steel can improve the environmental performance of their entire value chains, even in products other than renewable infrastructure.

Version 1:

Reviewer comments:

Reviewer #1

(Remarks to the Author)

Thank you for the opportunity to review the paper.

The main issues raised during the initial review have been resolved or clarified. The title of the revised version is more appropriate than the previous one. Furthermore, the authors have incorporated into their study the analysis of copper, hydrogen technologies, and transmission lines. They have also identified certain steel stocks that may not be easily accessible, as requested in the previous review.

On the other hand, the authors clearly explain that Al, Ni, Co, and Si are excluded from this study due to their limited contribution to the new renewable infrastructure. Finally, batteries have also been excluded from the study because, as the authors state, their primary use will be in the transport sector. However, it should also be noted that the transformation of the transport sector is indeed part of the energy transition. Beyond this minor clarification, I believe the authors have done a good job during the revision and that the article is ready for publication.

Reviewer #2

(Remarks to the Author)

The authors have made substantial improvements to the manuscript compared to the previous version, particularly by incorporating analyses on recycled copper from fossil infrastructure and secondary metal supply for electrolyzers and transmission networks. These additions significantly enhance the completeness and contribution of the study.

Nevertheless, several issues require further attention to improve clarity and accuracy:

Abstract:

(1) In the fourth sentence, the time frame for the "projected demand for the energy transition" should be explicitly stated.

(2) In the final sentence, the term "repurposing" should be reconsidered, as it may not accurately reflect the focus of the study, which centers on "recycling."

Introduction:

(3) It should be clarified that only 6 out of the 22 materials are compared with their projected energy transition demand.

(4) Lines 29–30: Including a similar comparative analysis for copper would help contextualize the subsequent sentence and clarify the rationale for focusing on steel and copper recycling.

Results:

(5) Line 45: The term "fossil" in parentheses does not align with energy transition technologies and may cause confusion.

(6) Lines 51–53: A reference to the figure presenting these results should be provided.

(7) Line 80: The title should also include the recycling of copper to accurately reflect the content.

(8) Figure 3: The y-axis labels in subplots (b) and (d) are difficult to read.

Discussion:

(9) Lines 245–248: The argument in this sentence is unclear. Regardless of the application, the use of recycled materials can displace primary metal production.

Other Comments:

(10) The formatting of numerical intervals (e.g., "[4.71–9.06] Gt" in Line 38) appears inconsistent with conventional style. Consider removing the square brackets for better adherence to standard practice.

(11) In Lines 208 and 209, the full term "aluminum" should be replaced with its abbreviation ("Al") for consistency with the rest of the text.

Version 2:

Reviewer comments:

Reviewer #2

(Remarks to the Author)

The authors have successfully addressed all of my remaining concerns. The revisions are sufficient, and I have no further comments.

Response to Reviewers for: NCOMMS-25-31469-T

Notes: Comments in blue | Replies in black | Actions in **bold** | Quoted text from manuscript in *italic*
Indicated page, line, figure, or reference numbers refer to the revised manuscript and/or supplemental information with changes highlighted.

Reviewer #1:

This paper titled “Digesting fossil infrastructure for cleaner energy transitions”, analyzes the stock of 22 materials present in fossil infrastructure, including: mines, wells, refineries, and pipelines for coal, oil, and natural gas. It considers the demand for 6 materials of the energy transition (concrete, steel, aluminum, copper, nickel, and silicon) for electricity generation applications (photovoltaic and wind). After this analysis, it focuses on the benefits of steel recycling, as it finds that the stock from fossil infrastructure is of the same order of magnitude as the demand of steel for the energy transition. It considers important aspects for recycling, such as the global capacity of Electric Arc Furnaces (EAFs), the location of fossil infrastructure and recycling plants, and the approximate time needed to process this infrastructure. In addition, it compares the externality costs of recycling versus primary production.

The paper is very thorough and provides valuable information in the context of energy transition. Nevertheless, several aspects should be considered. In order of criticality:

Thank you very much for your feedback and suggestions, which we thoroughly addressed in this review, as discussed in detail below.

1.1. First, the paper only considers wind and photovoltaic power as renewable infrastructure, overlooking a key aspect of the energy transition: storage. The main disadvantage of renewables compared to fossil fuels is that renewables consist of an intermittent flow, whereas fossil fuels are a stock. In other words, fossil fuels are available on demand, while renewables are not. The necessary solution is storage. The energy transition will not succeed without storage, and therefore this aspect should be considered. One way to store electricity is through batteries, and the most efficient type (Li-ion) contains significant amounts of cobalt. For this reason, the statement in lines 53–55 (“For some of the materials contained in current stocks, no demand for the energy transition was reported in [30] (e. g. for cobalt (Co); see Fig. 2b), or they are not typically contained in renewable energy 55 infrastructure”) is surprising. Cobalt is often used in superalloys for gas turbines (see Figure S4), so this element could be recovered and reused in batteries. Moreover, cobalt is a conflict metal, critical, and very scarce in nature, so its inclusion is important.

Thanks for pointing this out. As discussed more in-depth below, recycling the Co in current fossil infrastructure might not play a major role. However, we think Cu does deserve attention and, as explained in detail later in the document, have extended the calculations to account for the latter.

Indeed, Co is a critical metal and much needed in NMC-type Li-ion batteries (nickel manganese cobalt), which are currently still the main type used for mobile applications (phones, laptops, etc.) and electric vehicles. Yet, LFP alternatives (lithium iron phosphate), which avoid the use of Co (and Ni), are developing fast and are already used in >40% of electric vehicles as traction batteries [1]. This uncertain development makes it hard to predict the extent to which Co recovered from superalloys currently present in fossil infrastructure will find applications in energy storage. Furthermore, the projected demand for Co is orders of magnitude larger than the Co available at present in current fossil infrastructure. As such, it seems sensible to conclude that its recycling from fossil infrastructure will likely not play a major role.

Action: Added explanation for excluding Co to section 2.1, lines 67 to 73.

“Demand for cobalt is also not reported in [38]. The future demand is driven by battery deployment and projected to reach almost 500 kt per year in 2050 [41]. In fossil infrastructure, cobalt is used in

superalloys for gas turbines and can be recovered by leaching superalloy scrap [42, 43]. The total stock available in the current fossil infrastructure is 6.92 kt (median, see Fig. 2b), much lower than the expected cumulative demand for cobalt of 8.5 to 10.9 Mt from 2025 to 2050 (0.06 to 0.08% of demand). Therefore, recycling cobalt from fossil infrastructure might not play a major role, which is why we exclude it from further analysis.”

1.2. Second, the paper only considers the recycling of steel, but why does it ignore the recycling of aluminum and copper? Figure 2a shows that the stock of copper and the demand for the energy transition are close. In addition, this metal is experiencing a significant decline in ore grade, which is causing increasing impacts on its extraction and could compromise its supply in the future. Furthermore, it is a strategic metal, as it is not only used in energy infrastructure but is also essential for digitalization. This paper would be much more complete if the recycling of these metals were included or it should be better justified why these metals have not been considered.

Thanks for raising this point, which prompted us to extend the calculations to include copper, as described in detail next. We agree with the Reviewer that considering additional metals would increase completeness, provided they could contribute significantly to the demand induced by the energy transition. We focused on steel due to its high stock size compared to the expected demand of the energy transition. We agree with the Reviewer that a similar argument applies to copper. Copper recycling from fossil infrastructure could supply around one third of the expected cumulative demand (line 55), which is also significant. We, therefore, decided to include copper in our new analysis.

However, we still believe aluminium could be omitted, since its stock in fossil infrastructure is only 9% of the expected cumulative demand (line 55) and would, therefore, play a minor role in reducing primary aluminium production. Furthermore, most applications in renewable energy infrastructure require wrought aluminium (e.g., module frames, mounting systems, battery electrodes, cables), which cannot be produced with cast aluminium scrap. Wrought aluminium in fossil infrastructure accounts for 64% of the total Al stock, so the wrought Al stock in fossil infrastructure can only contribute around 6% to the Al demand of the energy transition (lines 77 to 78).

The new results show that copper recycling could save up to 85 Mt CO_{2eq} (4% of combined steel and copper recycling savings; SSP2-NPi scenario, Fig. E3 and E5) and 1.7 trillion USD of externality costs (25% of total savings) in total (Fig. 4). We further find that utilizing recycled copper in clean energy systems could significantly reduce the externality cost of manufacturing them; for example, the externality costs of onshore wind turbine manufacturing would be reduced by almost 900,000 USD/MW (-45% compared to using primary copper, see Fig. 5b). We now also analyze the implications of using recycled copper in electrolyzers and transmission networks. Since copper is not extensively used in electrolyzers, total externality costs are only minimally reduced (up to -2%), whereas transmission network externalities could be reduced significantly (83%) due to the heavy reliance on copper in cables and transformers (Fig. E11).

Action: We now include copper recycling capacity prospects and environmental impact reductions of copper recycling in section 2.2.2, lines 128 to 142:

“Regarding copper, the secondary production capacity was 12.7 Mt in 2023 and could reach more than 18.5 Mt in 2050 under the current growth rate (see SM sec. S3 for details). If the current copper stock in fossil infrastructure were recycled evenly until 2050, it would require less than 30% of the idle copper recycling capacity in any year and scenario (utilization factor of 0.83; see Fig. 3c). Overall, copper recycling capacities will be sufficient for absorbing the additional copper scrap from fossil infrastructure.

Copper recycling reduces environmental impact for all 20 LCA indicators when compared to primary production (Fig. 3d for nine LCA indicators and Fig. E5 and E6 for other indicators). In the SSP2-NPi scenario, the largest reductions are found in metal depletion, particulate matter formation, ozone depletion, and biodiversity loss (>99%, see Fig. 3d). Reductions in marine eutrophication (98–99%), ionising radiation (94–95%), water use (95–96%), and global warming potential (94–95%) are only

slightly smaller. These benefits arise mainly from avoiding copper mining and its sulfidic tailings—which release toxic compounds and drive eutrophication—as well as from avoided emissions of smelting (see Fig. E10). Over the period 2025–2050, the relative impacts of recycled copper decrease monotonically for 17 of the 20 indicators, primarily because declining ore grades make primary production progressively more burdensome (see Fig. S6a in the SM for details)."

We also extended the Methods to include copper (lines 350 to 361).

"The functional unit is 10.03 Mt ([5.94–18.33] Mt) of copper produced as the current globally weighted average and projected for the period 2025–2050. We model LCIs both for the pyrometallurgical (accounting for 80% of today's primary production; [82]) and hydrometallurgical route (20%) for primary copper production and the electrorefining route for secondary production (see sec. S5.2 in the SM for details).

Primary pyrometallurgical production consists of mining and beneficiation of copper ore concentrate, smelting, fire-refining, casting, and electrorefining. Hydrometallurgical copper production includes mining, crushing, and grinding of copper ore, leaching, solvent extraction, and electrowinning. For secondary copper production, we model scrap collection, transport, fire-refining, casting, and the electrowinning process. We also include a pessimistic scenario for secondary copper production—assuming low-grade copper scrap—, where scrap is smelted to increase purity before being fire-refined. For more details on system boundaries applied, assumptions, and data sources for process inputs and outputs, see the SM Section S5.2."

We further extended the original calculations to determine the environmental impact and externality cost savings from copper recycling and expanded the Figures 1, 3, 4, and 5, as well as sections S1, S3, S5, S6, S8, S9, and S10 in the supplementary materials (SM) accordingly. We further added the Figures E2, E5, E6, and E10 to show additional results for copper. For details on the system boundaries for copper production and underlying assumptions, see section S5.2 of the supplementary materials, where additional assumptions are laid out.

Regarding aluminium, we now include the justification for excluding it from further analysis in section 2.1, lines 74 to 79.

"We also exclude Al, Ni, and Si from further analysis, as recycling the stocks could at best contribute <10% of the demand generated by the energy transition (9%, 4%, and 0.001% respectively). The potential of Al stocks in the energy transition is even lower when considering that most clean energy applications require wrought-alloy Al (e.g. module frames, mounting systems, cables), which only accounts for 64% of the total Al in fossil infrastructure (thus only around 6% of the energy transition demand of Al could be supplied)."

1.3. Third, the paper only considers electrical technologies but ignores hydrogen infrastructure. Green hydrogen (produced via electrolysis) is key to decarbonizing applications that are difficult to electrify (such as high-temperature heat). It is also critical for the production of e-fuels, as acknowledged by the authors in lines 180–182 ("For example, the carbon footprint of green hydrogen, an energy carrier and fundamental molecule to activate the inert CO₂ in carbon capture and utilization routes [60], could drop by as much as 30%"). However, this infrastructure is not considered in the study. Therefore, the authors should explain why these technologies are not included in the study.

Thank you for this comment, which prompted us to perform additional calculations to cover this relevant point. We had initially excluded the assessment of electrolyzers, as many previous studies in the literature showed that the environmental impacts of hydrogen are dominated primarily by the impact of the electricity powering the electrolysis (i.e., 90–96% of climate impact dictated by the electricity source, and 58–88% in impacts of mineral resource scarcity [3]). Hence, making renewable electricity (i.e. predominantly solar and wind) greener with the use of repurposed steel and copper will also reduce the impacts of hydrogen produced using this electricity (sec. 2.4).

We agree that it is relevant to investigate further the potential role of repurposed steel and copper in the production of electrolyzers. To address this, we analyze in the new version three relevant electrolyzer technologies (proton-exchange membrane (PEM), alkaline electrolysis cells (AEC), and solid oxide electrolyzer cells (SOEC)), finding that the carbon footprint of the electrolyzer could be reduced by 3%, 39%, 35% respectively, when recycled steel and copper are used (Fig. E11a). Overall, using electrolyzers based on recycled metals from fossil infrastructure for green hydrogen production with solar energy, would reduce the carbon footprint of hydrogen by 1–12% depending on the electrolyzer (SM Fig. S13). Together with improvements in the footprint of solar infrastructure for electricity supply, this would translate into reductions in the carbon footprint of green hydrogen in the range 27–39% depending on the electrolyzer chosen (Fig. S13).

Action: We assess the climate impact and externality cost reduction of electrolyzer production when using recycled steel and copper from fossil infrastructure and display the results in the Extended results section, Figure E11. We now also refer to our assessment of electrolyzers in the main text, lines 225 to 231:

“Utilizing recycled steel and copper can also reduce climate impacts and externality costs of other relevant infrastructure, such as electrolyzers for green hydrogen production and power transmission infrastructure (see Fig. E11; details on the inventories in the SM sec. S5.6; we exclude batteries as their main application is expected to be primarily in the transport sector [38]). The alkaline electrolyzer shows the largest reduction for climate impact (38.8%) and externality cost (27.1%), both driven by recycled steel use. In contrast, benefits for power transmission infrastructure construction are driven by recycled copper use, leading to a 36.2% and 83.3% decrease in climate impact and externality cost, respectively.”

1.4. Fourth, most of the steel corresponds to oil and gas extraction equipment (line 43). However, much of this infrastructure is located in very remote areas or consists of pipelines at some depth, which could make metal extraction challenging. How is the dismantling of this infrastructure being approached?

Thanks for raising this point. Indeed, much of the steel in fossil infrastructure is tied to oil and gas extraction, where dismantling practices vary by location and accessibility. Offshore stocks represent, according to our findings, only ~2% of total steel (SM Fig. S1), with nearly half of the steel being in platforms where retrieval is common and feasible (as is done e.g., in the North Sea [4]). Offshore pipelines—around 1% of steel stocks—are often trenched on the seafloor and not recovered (SM sec. S11). Onshore pipelines—accounting for ~19% of steel—are technically retrievable, though often left buried due to cost unless regulations mandate removal (SM sec. S11). Likely inaccessible stocks include steel casing left in closed boreholes (~6% of total steel) and structural steel in underground coal mines (~4%), which may remain in place to avoid ground instability (SM Fig. S1, SM subsec. S11.1). Overall, ~11% of steel is estimated to be inaccessible for recycling (SM sec. S11).

Action: We have now added a comment on potentially inaccessible stocks in section S11 of the supplementary materials (limitations section). Limitations are referred to in line 283 of the discussion section in the main manuscript.

1.5. Fifth, to maximize the recycling of alloying metals in steel (some of which are critical, like Ni), it could be important to classify them in order to obtain steel of similar quality. Has this aspect been considered?

Thanks for this comment. This classification is indeed possible based on the data available, as discussed next. Most steel present in fossil infrastructure is construction steel (rebar, 94.6% to our findings, see SM Fig. S1). This type of steel is used for beams, supports, buildings, pipes, and casings. Specialized low-alloyed steel accounts for 3.8% and stainless steel (chromium steel) for 1.6% of the total steel in fossil infrastructure (see SM Fig. S1). The location of different types of alloys is well known in large assets like fossil infrastructure. Hence, we assume that sorting will be efficient,

allowing to recycle most of the steel types without a substantial loss of quality. For recycling, we model the corresponding recycling process for each of the three steel types assessed. Mounting systems for solar photovoltaic (PV), steel towers for wind turbines, and pylons for power lines all need predominantly construction steel, and could also use other low-alloyed steel, which matches the majority of the stock present in the fossil infrastructure (98.4%).

Action: We added a figure with the shares of steel by type to SM (section S1) and referred to it in line 189 of the main manuscript.

1.6. Lastly, regarding lines 108–109 (“Between 25 and 45 billion m²a of land could remain unoccupied”): does this figure account for the hectares that will be occupied by renewable technologies?

This number refers to the land occupation reduction in the material supply chains. We have clarified this further in the text, lines 147 to 149. Note that we now also include the avoided land occupation from copper recycling.

Action: Clarified the source of land occupation reduction in lines 147 to 149:

“Between 49 and 80 billion m²a of land could remain unoccupied from 2025 to 2050 due to the reduction in land required for steel and copper production (e.g. for mining, foresting), representing roughly the land occupation of Luxembourg [49].”

1.7. In conclusion, the title promises the digestion of fossil infrastructure for the energy transition but only focuses on electricity generation (wind and solar technologies) and steel recycling. Therefore, the authors should consider broadening the scope of the paper (including the recycling of more metals such as copper or cobalt and considering more technologies of the energy transition such as batteries or renewable hydrogen generation), or giving it a more specific title.

Thanks to your feedback and the positive assessment of our piece.

Prompted by the Reviewer, we have broadened the scope of our paper and performed additional calculations to address the main concerns raised. The new version includes recycling of copper, while discussing the potential inclusion of Al and Co, omitted here due to their expected much less relevant role in the same context. It also investigates the use of repurposed steel and copper in transmission infrastructure and electrolyzers via additional calculations, originally omitted in the previous version.

Our new results reinforce the finding that recycling fossil infrastructure can be a valuable strategy for reducing primary raw material demand for the energy transition (145% in steel and 32% in copper; line 46 and line 55) and mitigating environmental impacts, particularly when used for renewable energy technologies (up to 39% and 36% climate impact reduction for electrolyzers and transmission infrastructure, respectively; lines 229 and 231).

Besides all of these changes, since we heavily focus on recycling, and to avoid potential misinterpretation of the term ‘digesting’, we decided to propose a new title.

Action: Besides actions taken to address comments 1.2 and 1.3 (which address similar concerns), we change the title to “Recycling fossil infrastructure for cleaner energy transitions”.

Reviewer #2:

2.0. Thanks for your invitation. There are several major logical gaps in this study that warrant careful attention.

We thank the Reviewer for the thorough review of our manuscript and the many important and helpful comments. We have carefully addressed all the comments raised by the Reviewer, which have enhanced the quality of our manuscript. Specifically, we have substantially enlarged the scope of the analysis by including copper (and discussing the potential role of recycling cobalt and aluminum) and the use of recycled metals in electrolyzers and grid infrastructure.

2.1. First, while the study devotes significant effort to discussing recycled steel, its title focuses on “digesting fossil infrastructure,” which appears somewhat inconsistent with the article’s main content.

Thanks for pointing this out. We agree that our study has its focus on recycling materials from fossil infrastructure, which does not become apparent from reading the title. The term “digesting” could be unclear to readers. We therefore retitled our manuscript to increase clarity.

Action: Changed manuscript title to “*Recycling fossil infrastructure for cleaner energy transitions*”.

2.2. Second, recycled steel from decommissioned fossil infrastructure can be utilized across a variety of sectors, including construction and transportation, not necessarily in renewable infrastructure. Why, then, is the analysis limited to energy transition infrastructure? A clearer logical connection between the recycling of fossil infrastructure and the development of clean energy systems needs to be established.

Thanks for the comment. Indeed, recycled steel and other materials could be used across various sectors. However, the decommissioning of fossil energy infrastructure is contingent on its replacement by renewable energy systems. Our focus is, therefore, on how the energy transition can be supported through the recycling of the assets that become obsolete, as this enables the dismantling of fossil-based systems. We also note that the recycled material could be used in other applications (e.g., buildings, machinery; line 246), leading to equivalent impact reductions. In this sense, our results are general enough regardless of the final destination of the recycled material. For practical reasons, we limit the contextualization of the impact reduction values to uses in the energy infrastructure, but only to put them into perspective. We could have adjusted the impact values of other systems relying on recycled material, too.

However, when used for additional consumption instead of replacing non-recycled materials, recycled materials would overall not lower the environmental impact of society nor reduce supply constraints in the energy transition.

That said, we agree that this logical connection could be made more explicit and have clarified it in the second paragraph of the introduction.

Action: Clarified the connection between fossil infrastructure and the renewable energy system in lines 13 to 15.

“We focus here on how the energy transition can be supported by using recycled metals from fossil infrastructure in clean energy systems, as renewable energy systems are expected to replace fossil ones, thereby making the resulting waste from decommissioning them available for recycling purposes.”

We also come back to the point of using recycled materials for other purposes in the discussion, lines 245 to 248.

“Recycled materials may be redirected toward non-energy-related applications (e.g. buildings, machinery), but if they merely supplement overall consumption rather than replace primary materials, they will neither reduce society’s environmental impact nor alleviate resource constraints during the energy transition.”

2.3. Third, the study's definition of energy transition infrastructure seems to include only wind turbines and photovoltaic systems—why are other forms of clean energy infrastructure excluded?

Thank you very much for this question. A similar point was also brought up by Reviewer #1 in comment 1.3. In the original manuscript, we focused on solar and wind energy as they are currently expected to be the most scaled and deployed power technologies in many energy transition scenarios. We do agree, however, that other clean energy technologies, such as electrolyzers for green hydrogen production and power transmission networks, are also important. Hence, we now include them in our assessment. We exclude batteries from our assessment because their main application is expected to be in the transport sector, less so in the energy sector, as we now justify in line 227.

We find that replacing primary steel and copper with recycled materials in electrolyzers and transmission network infrastructure can reduce the carbon footprints by up to 39% and 36%, respectively (lines 229 and 231). Further, externality costs could also be reduced, more precisely, by between 2% and 27% depending on electrolyzer type (mainly driven by replacing primary steel) and by 83% for medium voltage grid infrastructure (mainly driven by replacing primary copper; Fig. E11).

Action: Performance of life cycle assessment on three electrolyzer types and power transmission infrastructure, and inclusion of results in the Extended results section, Figure E11. Reference to additional results and justification for excluding batteries added in line 225:

“Utilizing recycled steel and copper can also reduce climate impacts and externality costs of other relevant infrastructure, such as electrolyzers for green hydrogen production and power transmission infrastructure (see Fig. E11; details on the inventories in the SM sec. S5.6; we exclude batteries as their main application is expected to be primarily in the transport sector [38]). The alkaline electrolyzer shows the largest reduction for climate impact (38.8%) and externality cost (27.1%), both driven by recycled steel use. In contrast, benefits for power transmission infrastructure construction are driven by recycled copper use, leading to a 36.2% and 83.3% decrease in climate impact and externality cost, respectively.”

Details on system boundaries and data sources added to supplementary materials, section S5.2.

2.4. Fourth, even in the era of energy transition, it remains necessary to retain some fossil fuel infrastructure. Does this mean that the existing stock of fossil infrastructure does not need to be entirely “digested”?

We thank the Reviewer for the question. Indeed, this is true; not all fossil infrastructure might be recycled, as we clarify now in the manuscript. We are focusing on the *potential* of materials that can be recycled, as we point out in line 28. It is certainly likely that parts of the fossil infrastructure are retrofitted for carbon capture, or are targeted by other circular strategies such as rethinking or reusing (thus, changing the function without full dismantling, e.g., gas pipelines to a hydrogen pipeline). This will delay the available waste flow of materials, but will not reduce it, as even reused infrastructure will eventually be decommissioned.

However, around 11% of the existing steel stocks can be classified as inaccessible (also addressing comment 1.4; SM sec. S11). Offshore pipelines—around 1% of steel stocks—are often trenched on the seafloor and not recovered. Onshore pipelines—accounting for ~19% of steel—are technically retrievable, though often left buried due to cost unless regulations mandate removal (SM sec. S11). Likely inaccessible stocks include steel casing left in closed boreholes (~6% of total steel) and structural steel in underground coal mines (~4%), which may remain in place to avoid ground instability (SM subsec. S11.1). Overall, ~11% of steel is estimated to be inaccessible for recycling (SM sec. S11).

Action: Added discussion on the effect of other circular strategies on waste availability in section S11 of the SM:

“Recycling is one of several circular strategies that could be applied to obsolete infrastructures. Other strategies, such as reusing pipelines in district heating networks, retrofitting plants for carbon capture, or reusing power plant sites for grid storage infrastructure, could potentially avoid more costs and impacts or bring other advantages due to narrower cycles. Nevertheless, reusing will delay the available waste flow of materials, but will not reduce it, as even reused infrastructure will eventually be decommissioned. Future research can compare different circular strategies to find potential synergies or trade-offs.”

Added assessment of potentially inaccessible stocks in section S10 of the SM (limitations section):

“We are assessing the total potential of materials that could be recycled from fossil infrastructure, not accounting for material stocks in infrastructure that might be inaccessible. This might be the case for subsea offshore infrastructure (pipelines, oil and gas rigs), however, only 2% of steel stocks and less than 0.01% of copper stocks respectively are located offshore (see Fig. S1). Of these offshore stocks, 45% of steel and 98% of copper are stored in oil and gas wells, for which platform retrieval after decommissioning is common and feasible (e.g., in the North Sea; [43], although concrete jackets often remain). Offshore pipelines (around 1% of steel stocks) are often buried in the seafloor and not recovered [44].

Other fossil infrastructure might be inaccessible because it is buried, such as oil and gas borehole equipment, pipelines, and underground mines. In closed boreholes, part of the oil and gas borehole equipment (steel casing) is left permanently in the ground to stabilize the borehole, whereas in open hole and barefoot wells, steel casing is not always required [45]. Under the assumption that all decommissioned wells need steel casing, we estimate the steel kept in ground as being 12% (3–23%) of gas and oil well infrastructure (see SM section S11.1 for details), representing around 6% (2–11%) of the total steel stock in fossil infrastructure. This fraction of the steel stock might be unavailable for recycling.

Additionally, steel in underground coal mines is used for structural support and might lead to ground instability if removed. Leaving steel in underground coal mines in place would reduce the available steel stock in fossil infrastructure for recycling by another 4% (see Fig. S1).

In contrast, retrieval of onshore pipelines (18.6% and 0% of total global steel and copper stocks in fossil infrastructure) is technically feasible, although it might be uneconomical. Buried pipelines are allowed to be abandoned after use in some jurisdictions (e.g., in the US and the UK; [46, 47]). In cases where the economic value of the scrap does not compensate for the economic cost of retrieval, regulations would be needed that incentivize the removal of buried pipelines (or make removal mandatory) to ensure the availability of these stocks.

In total, the share of steel stocks that are inaccessible for recycling amounts to an estimated 11% (6–16%) and includes stocks in oil and gas well steel casings, underground mines, and offshore pipelines. Almost all copper stocks are likely accessible since they are not needed for structural purposes.”

Limitations are referred to in line 283 of the discussion section in the main manuscript.

Abstract

2.5. The first two sentences suggest that recycling can mitigate the environmental impacts of renewable energy infrastructure, which could make the readers think the topic of this study is about recycling renewable infrastructure. However, the focus of your study is actually on the recycling of fossil infrastructure. This creates a logical inconsistency that may confuse readers.

Thank you for pointing that out. We agree that the sentences can be misunderstood as referring to renewable energy infrastructure recycling. We now clarify that we refer to recycling materials in urban mines for sourcing the required materials for renewable energy infrastructure.

Action: We rephrased the second sentence in the abstract as follows:

“Sourcing these materials by recycling material from urban mines can mitigate such impacts, but the potential of recycling depends on waste availability.”

2.6. Regarding the 1.35 Gt steel stock mentioned in the abstract—what is its source? Does it refer to the in-use stock within fossil infrastructure to be obsolete? Additionally, where does the projected demand for the energy transition come from, and what is the time frame it covers? I think you need to clarify clearly the sources of material demand for energy transitions throughout the whole paper.

We now clarify that the steel stock refers to the *in-use stock of fossil infrastructure*, based on our own analysis. The projected steel demand for the energy transition is sourced from [5], as stated in the caption of Figure 2. The time frame for this demand spans 2020 to 2050, as indicated in line 46 and the caption of Figure 2. We have also clarified in lines 45–46 that the demand covers technologies including solar, wind, nuclear, fossil, biomass, hydro, and geothermal energy systems, as well as transmission and distribution infrastructure.

Action: Specified focus on in-use stocks in line 27:

“To fill this gap, we map global in-use stocks in the current fossil infrastructure for 22 materials, quantifying the recycling potential and comparing it with estimated materials demand induced by transition pathways”

Changed time frame of demand from “until 2050” to “2020 to 2050” in line 46. Clarified technologies included in the material demand projections of energy transitions in lines 45 and 46:

“Steel is the second largest stock, amounting to 1.34 Gt ([0.84–2.26] Gt), matching cumulative steel demand estimates for energy transitions including energy generation (solar, wind, nuclear, fossil, biomass, hydro, and geothermal energy), transmission, and distribution infrastructure from 2020 to 2050 (145% of the median demand, 0.93 Gt ([0.51–2.85] Gt) depending on the scenario [38], see Fig. 2a; material demand of grid-scale batteries is excluded since battery capacity in the transport sector (electric vehicles) may be leveraged [38])”

2.7. What is the time period over which the reported CO₂ emissions reductions and cost savings are calculated?

We report emissions reductions and cost savings for different possible years of recycling from 2025 to 2050 (line 145), assuming that the total steel and copper stock is recycled within this time frame.

Action: We clarified the time frame considered in the section on cumulative impact (line 144):

“Scaling environmental benefits to the total steel and copper contained in fossil infrastructure (investigating a time frame of 2025–2050),...”

We further clarified the time frame considered in the section on cost savings (lines 175 to 178):

“Recycling fossil steel and copper stocks could avoid externality costs between 4.18 and 7.00 trillion USD₂₀₂₃ (Fig. 4a, SSP2-NPi scenario, 75–78% from steel recycling; between 4.05 and 11.69 trillion USD₂₀₂₃ across investigated SSP scenarios, Fig. E1 and E2), depending on the year of intervention (2025–2050).”

2.8. The final sentence references 'socioeconomic and geopolitical issues,' yet these aspects are not addressed or supported by the findings presented in this study.

While we briefly mention potential socioeconomic and geopolitical implications in the discussion, we believe that this topic can be omitted in the abstract. Therefore, we have removed the sentence from the abstract.

Action: Removed mention of socioeconomic and geopolitical implications from the abstract.

Introduction

2.9. Line 26: the steel stock is (1.34 Gt) is a little different from the figure (1.35 Gt) in abstract.

We thank the Reviewer for pointing out this issue, which we have amended.

Action: Harmonized steel stock size to the correct 1.34 Gt throughout the manuscript.

2.10. Figure 1: what is 'soon-to-be-obsolete' fossil infrastructure? How to determine which infrastructure is about to be obsolete?

We were referring to fossil infrastructure that might be phased out in the energy transition, and, therefore, might not be used anymore. Under further consideration, we see that 'soon-to-be-obsolete' is ambiguous, and therefore decided to remove the adjective.

Action: Removed 'soon-to-be-obsolete' in the caption of figure 1.

Results

2.11. Line 43-46: specific figures or tables need to be depicted to support these results, which can not be seen from Figure 2.

The Reviewer is right that the figures on the steel stocks by infrastructure type are not visible in Figure 2. We, therefore, added a figure to the supplementary material and referred to it in the main manuscript.

Action: Added figure with steel and copper stock by fossil infrastructure type to supplementary material (Fig. S1). Added reference to figure in line 50.

2.12. Figure 2a: There are only 6 materials shown in Figure 2a, not 22.

The Reviewer is completely right: the whole figure depicts 22 materials, but subplot "a" only depicts six materials. We amended the number accordingly.

Action: Changed to "six materials" in the caption of figure 2.

2.13. Figure 2b: Again, you need to define clearly the material demand for energy transitions. Actually, cobalt is a critical material used in batteries, which are also important for energy transitions, but you didn't take cobalt as an energy transition metal in this study.

Indeed, cobalt is used in NMC-type batteries mostly for mobile applications today. LFP-type batteries are emerging and may potentially replace the currently dominating NMC types [1]. In LFP, no cobalt is needed. Yet, as this substitution is uncertain and IEA projects the demand for cobalt to double by 2050, we included a potential analysis of cobalt recovered from super alloys currently embodied in fossil infrastructure. Though technically possible, recovering all cobalt from the fossil infrastructure can at best contribute 10 kt of cobalt in total, which is much smaller than the current annual demand (280 kt/a) and the projected demand for 2050 (450 kt/a). Hence, cobalt recycling from fossil infrastructure may not play a significant role in reducing the impacts from batteries. We have included this discussion in the paper.

Action: Included an analysis of Co in section 2.1 and justified its exclusion from further analysis (lines 67 to 73):

"Demand for cobalt is also not reported in [38]. The future demand is driven by battery deployment and projected to reach almost 500 kt per year in 2050 [41]. In fossil infrastructure, cobalt is used in superalloys for gas turbines and can be recovered by leaching superalloy scrap [42, 43]. The total stock available in the current fossil infrastructure is 6.92 kt (median, see Fig. 2b), much lower than the expected cumulative demand for cobalt of 8.5 to 10.9 Mt from 2025 to 2050 (0.06 to 0.08% of demand). Therefore, recycling cobalt from fossil infrastructure might not play a major role, which is why we exclude it from further analysis."

2.14. Line 61: I think there needs a full explanation of why you chose steel to investigate the recycling benefits. The “relevance” here is too ambiguous.

The original focus on steel is based on our findings mapping material stocks in fossil infrastructure. As Figure 2 shows, the steel stock is the only material stock in fossil infrastructure that closely matches the steel demand induced by the energy transition. For all other materials, demand in the energy transition is substantially larger than the corresponding availability in the fossil infrastructure.

Yet, following this and Reviewer #1’s comments, we have expanded the detailed analysis to copper and clarified further the selection of materials we consider for a detailed recycling analysis. We now include materials that are stored in the fossil infrastructure, which supply more than 10% of the energy transition demand (145% for steel and 32% for copper; lines 46 and 55).

Action: Explanation of material selection for further analysis is expanded in section 2.1, lines 67 to 79.

“Demand for cobalt is also not reported in [38]. The future demand is driven by battery deployment and projected to reach almost 500 kt per year in 2050 [41]. In fossil infrastructure, cobalt is used in superalloys for gas turbines and can be recovered by leaching superalloy scrap [42, 43]. The total stock available in the current fossil infrastructure is 6.92 kt (median, see Fig. 2b), much lower than the expected cumulative demand for cobalt of 8.5 to 10.9 Mt from 2025 to 2050 (0.06 to 0.08% of demand). Therefore, recycling cobalt from fossil infrastructure might not play a major role, which is why we exclude it from further analysis.

We also exclude Al, Ni, and Si from further analysis, as recycling the stocks could at best contribute <10% of the demand generated by the energy transition (9%, 4%, and 0.001% respectively). The potential of Al stocks in the energy transition is even lower when considering that most clean energy applications require wrought-alloy Al (e.g. module frames, mounting systems, cables), which only accounts for 64% of the total Al in fossil infrastructure (thus only around 6% of the energy transition demand of Al could be supplied).”

2.15. Line 67-68: how did you conclude it?

We arrived at the years needed to recycle the steel stock in the original manuscript by dividing the potential steel stock available (upper and lower estimates) by the current idle recycling capacity. However, as this is a static value, we changed the approach in the new version of the manuscript to a different approach (see reply to comment 2.16).

Action: See action taken for comment 2.16.

2.16. Line 65: What does “Mt/a” mean—does it refer to “Mt per annum”? If so, comparing the ideal recycling capacity (in Mt/a) with the total steel stock is conceptually misleading, as the stock will not retire in a single year but over an extended period. A more appropriate comparison would be between the recycling capacity and the annual volume of steel retiring from fossil infrastructure.

Yes, “Mt/a” does indeed mean “Mt per annum” and refers to the global capacity of steel that can be recycled globally within one year.

Our assessment aims to assess whether or not recycling capacity may become a limiting factor for using the materials recycled from the fossil infrastructure. We showed in the original manuscript that the entire fossil steel stock can be recycled in a few years with the already existing and currently idle recycling capacity. We understand that this approach can be misinterpreted since—as the Reviewer mentions—the whole stock is not retiring in a single year and the idle recycling capacity might change in the future (comment 2.17).

We, therefore, conducted additional calculations with refined assumptions, finding that the new results lead to similar conclusions. First, instead of relying on current recycling capacity, we now use projections of recycling capacities until 2050. Moreover, we now assume an annual waste volume

stemming from the stock retiring evenly over the period 2025 to 2050. A constant retirement rate of fossil infrastructure until 2050 translates into a linear decline of fossil infrastructure stocks until reaching net zero in 2050.

Following this refined analysis, it becomes apparent that in any year and scenario, the additional steel waste flow utilizes less than 74% of the idle steel recycling capacity in any year and scenario (line 91). For copper, which we now include, the additional copper scrap volume utilizes less than 30% of the idle recycling capacity in any year 2025 and 2050 (line 130). Thus, the recycling capacity might be enough to absorb the steel and copper stock from fossil infrastructure.

Action: Changed Figure 3 and included a dynamic approach in section 2.2, with results for steel described in lines 90 to 93:

“If the steel scrap from fossil infrastructure were released evenly across the period 2025–2050, the annual additional steel scrap volume would then utilize less than 40% of the global idle recycling capacity in any year (see Fig. 3a, SSP2-NPi, never exceeds 74% in any other scenario, see SM Fig. S2), thus likely not posing a constraint on absorbing the additional steel waste.”

For copper in lines 129 to 132:

“If the current copper stock in fossil infrastructure were recycled evenly until 2050, it would require less than 30% of the idle copper recycling capacity in any year and scenario (utilization factor of 0.83; see Fig. 3c). Overall, copper recycling capacities will be sufficient for absorbing the additional copper scrap from fossil infrastructure.”

Added recycling capacity projections to the SM section S3.

2.17. Line 71-77: How did you tell “recycling of steel from fossil infrastructure would take five to 14 years with existing idle recycling capacity” from Fig. 3a? I don’t see any information about recycling time in Fig. 3a. In addition, the definition of recycling time needs to be explained cautiously because the recycling capacity can increase, which will shorten the recycling time. In addition, it is confused and odd that you evaluate the feasibility of recycling steel stocks through a comparison between the recycling time and energy transition time. I think a better story should be told here.

As we highlighted in our response to comment 2.16, we have now refined the calculations, considering varying recycling capacities and an annualized release of waste from fossil infrastructure. We show that the annual steel and copper waste released from fossil infrastructure never exceeds the respective idle recycling capacity in the period 2025 to 2050, therefore being sufficient for absorbing this additional waste stock.

Action: See actions taken to address comment 2.16.

2.18. Figure 5: I understand that the aluminum is replaced by secondary steel so that reducing the carbon emissions related to aluminum. I suggest that you can separate the carbon emissions of steel in PV systems in the right bars (hatched areas) into two segments correspond to the recycled steel that replaces primary steel and the recycled steel that replaces aluminum, respectively.

We thank the Reviewer for this good suggestion. We agree that separating the contribution of recycling steel replacing primary steel, and recycling steel replacing aluminium enhances the clarity of the figure. We, therefore, modified Figure 5 so that the contribution of steel replacing aluminium is distinct from the contribution of the recycled steel that replaces primary steel.

Action: Adjustments to Figure 5.

2.18. Will it increase the installation costs by replacing Al with steel in PV mounting system? If so, the way replaces Al with steel maybe not economically viable in practice.

Indeed, we think steel mounting systems might influence the installation costs of PV due to two main reasons. First, steel mounting systems might be less expensive than aluminium mounting systems,

reducing material costs. Secondly, installing steel mounting systems might take more time, thus increasing labor costs.

Regarding material costs, galvanized steel profiles are less expensive per kilogram compared to aluminium profiles, thus we expect material costs to be slightly lower when using steel mounting systems (decrease by 50(21–84) USD/kWp or 3.0(1.2–5.0)% of the total installation cost; sec. S7 in the SM).

Even under the assumption that the steel mounting system doubles the labor costs for mounting, this would increase installation costs only marginally by 21(8–47) USD/kWp or 1.2(0.5–2.8)% of the total installation cost. The net cost savings of PV using steel mounting systems might, thus, be 28(–9–66) USD/kWp or 1.7(–0.5–3.2)% of the total installation cost. This estimated cost benefit should be taken with a pinch of salt in light of the lack of data collected from industry. Overall, PV installation costs might remain similar regardless of the type of mounting system used.

Action: Added cost analysis for PV installation with steel and aluminium mounting systems in section S7 in the supplementary materials.

Discussion

2.19. Line 178-183: All the benefits are actually attributed to the recycled steel. It is obvious that using recycled steel can improve the environmental performance of their entire value chains, even in products other than renewable infrastructure.

We appreciate the reviewer's observation and agree that recycled steel and copper is widely recognized for its environmental advantages. However, our study goes beyond reiterating known benefits. The novelty lies in quantifying the untapped potential of steel and copper embedded in fossil infrastructure and demonstrating its strategic relevance for the energy transition. We think that our work goes beyond previous studies in various ways:

(i) We assess the feasibility of mobilizing steel and copper stocks, showing that global idle recycling capacity is likely to be sufficient to absorb the released scrap without bottlenecks. (ii) We quantify the system-wide environmental and cost benefits of using this specific waste stream—including up to 1.95 Gt CO_{2eq} avoided (line 145) and 11.69 trillion USD (line 177) in externality cost savings—across multiple climate scenarios and years. (iii) We demonstrate how recycled steel from fossil infrastructure can enable the construction of up to 45 TWp of solar PV and 10 TWp of wind capacity (Table 1), exceeding global targets, and reducing the carbon footprint of these technologies by up to 41% (line 212).

Action: The main novelty was higher stressed throughout the manuscript.

Lines 90 to 93:

“If the steel scrap from fossil infrastructure were released evenly across the period 2025–2050, the annual additional steel scrap volume would then utilize less than 40% of the global idle recycling capacity in any year (see Fig. 3a, SSP2-NPi, never exceeds 74% in any other scenario, see SM Fig. S2), thus likely not posing a constraint on absorbing the additional steel waste.”

Lines 131 to 132:

“Overall, copper recycling capacities will be sufficient for absorbing the additional copper scrap from fossil infrastructure.”

Lines 221 to 224:

“Yet, overall, recycling steel and copper from fossil infrastructure could contribute significantly to the steel and copper demand of the energy transition and lower the environmental impacts and externalities costs of transitioning to solar and wind energy.”

References

- [1] Adamas Inside. LFP now commands 40 % of the global EV battery market. <https://www.adamasintel.com/lfp-now-commands-40-of-the-global-ev-battery-market/> (2025)
- [2] Global Critical Minerals Outlook 2025. (International Renewable Energy Agency, 2025). <https://www.iea.org/reports/global-critical-minerals-outlook-2025>.
- [3] Zhao, G., Kraglund, M. R., Frandsen, H. L., Wulff, A. C., Jensen, S. H., Chen, M. & Graves, C. R. Life cycle assessment of H₂O electrolysis technologies. *International Journal of Hydrogen Energy* 45, 23765–23781 (2020).
- [4] OSPAR Commission OSPAR Decision 98/3 on the Disposal of Disused Offshore Installations (as amended by OSPAR Decision 2024/01): Consolidated Text In OSPAR Commission Decisions and Agreements (2024). <https://www.ospar.org/documents?v=57705>.
- [5] Wang, S. et al. Future demand for electricity generation materials under different climate mitigation scenarios. *Joule* 7, 309–332 (2023).

Response to Reviewers for: NCOMMS-25-31469A

Notes: Comments in **blue** | Replies in black | Actions in **bold** | Quoted text from manuscript in *italic*
Indicated page, line, figure, or reference numbers refer to the revised manuscript and/or supplemental information with changes highlighted.

Reviewer #1:

1.0. Thank you for the opportunity to review the paper.

The main issues raised during the initial review have been resolved or clarified. The title of the revised version is more appropriate than the previous one. Furthermore, the authors have incorporated into their study the analysis of copper, hydrogen technologies, and transmission lines. They have also identified certain steel stocks that may not be easily accessible, as requested in the previous review. On the other hand, the authors clearly explain that Al, Ni, Co, and Si are excluded from this study due to their limited contribution to the new renewable infrastructure. Finally, batteries have also been excluded from the study because, as the authors state, their primary use will be in the transport sector.

We thank the Reviewer for the careful assessment and the positive feedback. We are pleased that our revisions have addressed the main issues to the Reviewer's satisfaction. We appreciate the Reviewer's recognition of these clarifications and believe that the revised manuscript now presents a more coherent and targeted analysis.

1.1. However, it should also be noted that the transformation of the transport sector is indeed part of the energy transition. Beyond this minor clarification, I believe the authors have done a good job during the revision and that the article is ready for publication.

We agree with the Reviewer that the transformation of the transport sector is an integral part of the broader energy transition, namely the shift from fossil fuels as the dominant energy carrier to alternative fuels and electricity. We recognize that our initial justification for excluding batteries (line 49) could be misinterpreted as suggesting that the transport transition is not part of the energy transition. To avoid this misunderstanding, we have clarified the relation between the energy transition and the transport transition.

We would like to thank the reviewer once again for the insightful comments, which have greatly enhanced the rigor, conciseness, comprehensiveness, and clarity of the revised manuscript.

Action: Clarified that the transformation of the transport sector is part of the energy transition in lines 287 to 289:

“Further, for simplicity, we do not focus on recycled materials’ utilization in the transport sector (e.g. electric vehicles); nevertheless, the transport transition is also integral to the broader energy transition (e.g. energy storage in battery-electric vehicles).”

Reviewer #2:

2.0. The authors have made substantial improvements to the manuscript compared to the previous version, particularly by incorporating analyses on recycled copper from fossil infrastructure and secondary metal supply for electrolyzers and transmission networks. These additions significantly enhance the completeness and contribution of the study.

We thank the Reviewer for this positive assessment of our previous revision and for the additional comments and suggestions, which we are happy to address below.

Abstract:

2.1. Nevertheless, several issues require further attention to improve clarity and accuracy:

In the fourth sentence, the time frame for the "projected demand for the energy transition" should be explicitly stated.

We agree with the reviewer that adding the time frame would increase clarity, and we, therefore, added it to the abstract.

Action: Added demand projection time frame (2020 to 2050) to the abstract.

"Among many materials in fossil infrastructure, recycling steel and copper is particularly appealing, as their stocks (1.34 Gt and 10.03 Mt) align with the projected demands for the energy transition from 2020 to 2050 (145% and 32% of median demand, respectively)."

2.2. In the final sentence, the term "repurposing" should be reconsidered, as it may not accurately reflect the focus of the study, which centers on "recycling."

Thanks for pointing this out. Indeed, the term "repurposing" reflects several circular strategies, which is why we have now replaced it with "recycling" in the abstract and throughout the revised manuscript where it appeared in the same context (line 238) or with "retiring" where it appeared in the context of decommissioning (line 265).

Action: Replaced "repurposing" with "recycling" where it appeared in the context of recycling steel and copper from fossil infrastructure and with "retiring" where it appears in the context of decommissioning.

Abstract:

"These findings provide strong evidence of the benefits of recycling fossil infrastructure and underscore the need for policies to expedite the energy transition."

Line 238:

"While adopting greener production routes is often hindered by higher costs [61–63], recycling steel and copper from fossil infrastructure could represent a win-win alternative."

Line 265:

"Resistance to retiring may vary depending on the type of infrastructure."

Introduction:

2.3. It should be clarified that only 6 out of the 22 materials are compared with their projected energy transition demand.

We recognize that the sentence in lines 27–29 could be misinterpreted as suggesting that all 22 materials were compared with their projected energy transition demand. Indeed, only six materials are benchmarked against their energy transition demand. We have revised the sentence to eliminate this ambiguity and ensure the distinction is clear.

Action: Clarified sentence in lines 27–29.

“To fill this gap, we map global in-use stocks in the current fossil infrastructure for 22 materials, quantifying the recycling potential, and comparing the stocks of six materials with estimated material demand induced by transition pathways.”

2.4. Lines 29–30: Including a similar comparative analysis for copper would help contextualize the subsequent sentence and clarify the rationale for focusing on steel and copper recycling.

The Reviewer is right, this paragraph (lines 29–31) refers to our results on steel recycling, while not mentioning copper, which is now included in the further analysis. We, therefore, added a comparative analysis for copper.

Action: Added a contextualized reference to the findings regarding copper stocks in lines 31–32:

“We find that steel is the largest metal stock in fossil infrastructure (1.34 Gt) and is within the range of estimated demand in transition pathways aiming at 1.5 °C and 2 °C heating [38]. Copper is another promising material to recycle (10.03 Mt in fossil infrastructure), which could contribute around one third of the median energy transition demand.”

Results:

2.5. Line 45: The term "fossil" in parentheses does not align with energy transition technologies and may cause confusion.

Thank you for spotting this. Fossil energy generation is, of course, not included in the clean transition technologies.

Action: Removed “fossil” from the list in parentheses (line 47).

“Steel is the second largest stock, amounting to 1.34 Gt (0.84–2.26 Gt), matching cumulative steel demand estimates for energy transitions including energy generation (solar, wind, nuclear, biomass, hydro, and geothermal energy),...”

2.6. Lines 51–53: A reference to the figure presenting these results should be provided.

We now include a reference to Figure 3a when listing the countries with the largest steel stocks in fossil infrastructure (line 54).

Action: Included a reference to Figure 3a in line 54.

“The five countries with the largest fossil steel stocks are China, Saudi Arabia, Russia, the United States, and Iran, with 164, 155, 152, 150, and 94 Mt steel respectively (see Fig. 3a),...”

2.7. Line 80: The title should also include the recycling of copper to accurately reflect the content.

The Reviewer is completely right. Due to editorial policies, we had to remove the level of subsections and therefore split the subsection 2.2 into three subsections, one for “Feasibility of recycling steel stocks and associated environmental impacts”; one for “Feasibility of recycling copper stocks and associated environmental impacts”; and one for “Total environmental benefits”.

Action: Split the former subsection 2.2 (line 85) into three subsections:

Line 85

“Feasibility of recycling steel stocks and associated environmental benefits”

Line 128

“Feasibility of recycling copper stocks and associated environmental benefits”

Line 144

“Total environmental benefits”

2.8 Figure 3: The y-axis labels in subplots (b) and (d) are difficult to read.

We see that the y-axis labels in Figure 3b and d might be difficult to read, especially the subscripts. Therefore, we replaced the labels with variants without subscripts.

Action: Replaced y-axis labels in Figure 3b and d.

Figure 3b:

“secondary steel impact as fraction of primary impact [-]”

Figure 3d:

“secondary copper impact as fraction of primary impact [-]”

Discussion:

2.9 Lines 245–248: The argument in this sentence is unclear. Regardless of the application, the use of recycled materials can displace primary metal production.

In our original argument, we focused on the energy sector, but certainly this does not imply that displacement only matters in energy-related contexts. In reality, displacement of primary production through secondary production is beneficial regardless of the sector.

We rephrased the sentence to avoid confusion.

Action: Rephrased sentence in line 247.

“We focus on the application of recycled materials in the energy sector, as fossil technologies need to be replaced with renewable ones. However, we clarify that the benefits of recycling materials can be realized regardless of the application sector as long as they displace primary production.”

Other Comments:

2.10. The formatting of numerical intervals (e.g., “[4.71–9.06] Gt” in Line 38) appears inconsistent with conventional style. Consider removing the square brackets for better adherence to standard practice.

Thank you for pointing this out. We changed the formatting numerical intervals to the more conventional style without square brackets (e.g. 1.34 Gt (0.84–2.26 Gt)).

Action: Removed square brackets in intervals across the main manuscript and supplementary materials.

2.11. In Lines 208 and 209, the full term “aluminum” should be replaced with its abbreviation (“Al”) for consistency with the rest of the text.

Thank you very much for pointing out this inconsistency. We replaced “aluminum” with its abbreviation in these occurrences.

Action: Replaced “aluminum” with its abbreviation “Al” after its first appearance in the manuscript (line 209 and line 210):

“Similarly, while steel is currently used in solar PV systems only for screws and hooks in rooftop installations and for support pylons in open-ground installations, it could replace commonly used Al for mounting systems [56] (for details see the SM sec. S8). Replacing Al in mounting systems for solar PV with secondary steel and using recycled copper in cables would curb greenhouse gas emissions by 39.2% and 24.8% for open-ground and slanted roof PV system production, respectively.”